# Mechanism and preclinical prevention of increased breast cancer risk caused by pregnancy

Svasti Haricharan[1,2†], Jie Dong[2†], Sarah Hein[1,2], Jay P Reddy[1,2], Zhijun Du[2], Michael Toneff[1,2], Kimberly Holloway[2], Susan G Hilsenbeck[2], Shixia Huang[1], Rachel Atkinson[3], Wendy Woodward[4], Sonali Jindal[5,6], Virginia F Borges[5,6], Carolina Gutierrez[2], Hong Zhang[7], Pepper J Schedin[5,6,8], C Kent Osborne[2], David J Tweardy[9], Yi Li[1,2]*

[1]Department of Molecular and Cellular Biology, Baylor College of Medicine, Houston, United States; [2]Lester and Sue Smith Breast Center, Baylor College of Medicine, Houston, United States; [3]Department of Clinical Cancer Prevention, MD Anderson Cancer Center, Houston, United States; [4]Division of Radiation Oncology, MD Anderson Cancer Center, Houston, United States; [5]Division of Medical Oncology, University of Colorado Denver Anschutz Medical Campus, Aurora, United States; [6]Young Women's Breast Cancer Translational Program, University of Colorado Denver Anschutz Medical Campus, Aurora, United States; [7]Department of Pathology, MD Anderson Cancer Center, Houston, United States; [8]Program in Cancer Biology, University of Colorado Denver Anschutz Medical Campus, Aurora, United States; [9]Department of Medicine, Baylor College of Medicine, Houston, United States

*For correspondence: liyi@bcm.edu

†These authors contributed equally to this work

Competing interests: The authors declare that no competing interests exist.

**Abstract** While a first pregnancy before age 22 lowers breast cancer risk, a pregnancy after age 35 significantly increases life-long breast cancer risk. Pregnancy causes several changes to the normal breast that raise barriers to transformation, but how pregnancy can also increase cancer risk remains unclear. We show in mice that pregnancy has different effects on the few early lesions that have already developed in the otherwise normal breast—it causes apoptosis evasion and accelerated progression to cancer. The apoptosis evasion is due to the normally tightly controlled STAT5 signaling going astray—these precancerous cells activate STAT5 in response to pregnancy/lactation hormones and maintain STAT5 activation even during involution, thus preventing the apoptosis normally initiated by oncoprotein and involution. Short-term anti-STAT5 treatment of lactation-completed mice bearing early lesions eliminates the increased risk after a pregnancy. This chemoprevention strategy has important implications for preventing increased human breast cancer risk caused by pregnancy.

## Introduction

Epidemiological studies have demonstrated that a first full-term pregnancy before age 22 greatly reduces breast cancer risk, but a pregnancy after age 35 is stimulatory (*MacMahon et al., 1970*; *Polyak, 2006*; *Schedin, 2006*). In the global context of increasing age at first pregnancy, it has become critical to identify the molecular mechanism underlying increased long-term breast cancer risk in parous women who had a late-age first pregnancy, so that prevention strategies may be developed to reduce this risk. When a very young woman becomes pregnant, her breast epithelia are unlikely to have accumulated cells with oncogenic mutations (*Crowley and Curtis, 1963*; *Nielsen et al., 1987*; *Lynch, 2010*). Normal breast epithelia, following extensive remodeling by a pregnancy,

**eLife digest** Pregnancy changes the probability that a woman will later develop breast cancer. If a woman's first pregnancy occurs before her 22nd birthday, the chances of developing breast cancer are reduced. However, if the first pregnancy occurs after her 35th birthday, there is an increased risk of breast cancer. It is not clear why this age-related difference exists, but as more women wait until their 30s to start a family, there is greater urgency to understand this difference.

Breasts undergo extensive changes during pregnancy. This remodeling makes their cells less likely to multiply, and also less likely to develop tumors, which could explain the protective effect of pregnancy for younger women. But why would older women not reap the same benefits? One hypothesis is that older first-time mothers are more likely than younger first-time mothers to already have breast tissue with cells carrying cancer-causing mutations, or to have clusters of abnormal precancerous cells.

Now, Haricharan et al. have tested this hypothesis by inserting two cancer-causing genes into female mice. Half of the mice were then made pregnant and allowed to nurse their young, whilst the other half were never mated. Although, both groups of mice later developed tumors, the mice that had been pregnant developed more tumors and did so faster.

The increased cancer levels in the mice that had been pregnant were not due to them having more precancerous cells at the early stages of pregnancy than the unmated mice of the same age. Further, the precancerous cells in the impregnated mice did not proliferate faster than those in the mice that were never pregnant. Instead, pregnancy weakened the protective process that culls pre-existing precancerous cells. These cells evaded destruction by activating a signaling pathway called the STAT5 pathway in response to pregnancy hormones.

Haricharan et al. also examined tissue samples from women with a very early form of breast cancer and found elevated levels of STAT5 in tumors from women who had been pregnant compared to those who had not been pregnant.

The good news is that precancerous cells do not always become cancerous. However, for those women with a high risk of developing breast cancer, Haricharan et al. suggest that temporarily reducing STAT5 activity after pregnancy with medication might reduce this risk. Treating mice with anti-STAT5 drugs for a few weeks after they finished nursing their young lessened the elevated cancer risk, and so the next challenge is to see if this approach will also be effective in human clinical trials.

have been reported to become more differentiated, less proliferative, and indolent to tumorigenesis (*Medina, 2004*). However, why a first pregnancy at an older age stimulates life-long breast cancer incidence remains mysterious. As a woman reaches the age of 35 or older, her breast epithelia are more likely to have accumulated cells with oncogenic mutations and to harbor precancerous early lesions than a woman in her 20s, based on autopsy studies (*Bartow et al., 1987*; *Nielsen et al., 1987*; *Welch and Black, 1997*). The effects of a pregnancy on these preexisting cancer-precursor cells have not been rigorously tested. If pregnancy instigates cancer evolution from these premalignant cells, the result may explain the age-dependent impact of pregnancy on breast cancer.

Using conventional transgenic mouse models to address the effect of pregnancy on preexisting mammary cells has been problematic because the transgenic promoter, and thus the oncogene it regulates, is usually dramatically induced by pregnancy and lactation hormones and also because transgenic oncogenes often impair the normal development of the mammary gland (*Vargo-Gogola and Rosen, 2007*). We have reported intraductal injection of a Rous sarcoma virus-based vector, RCAS, as a means of introducing oncogenes into a small subset of cells in the normally developed mammary gland that has been made susceptible to infection by transgenic expression of the gene encoding the RCAS receptor TVA from the MMTV promoter (MMTV-*tva*) (*Du et al., 2006*). The transgenic *tva* is only required for the initial infection, while the oncogene is transcriptionally controlled by the proviral RCAS LTR, which is constitutive and not influenced by reproductive hormones (*Toneff et al., 2010*; *Li et al., 2011*). Using this method, we tested here the effect of a single pregnancy on carcinogenesis from a small number of oncogene-activated mammary cells in the context of normal mammary epithelia and explored the underlying mechanism of, and means to prevent, increased long-term breast cancer risk caused by a pregnancy.

## Results

### A full-term pregnancy promotes tumorigenesis from preexisting mutated breast cells

We chose *Erbb2* and *Wnt1* as the initiating oncogenes, because their gene products activate two distinct pathways that are frequently altered in human breast cancer (*Klaus and Birchmeier, 2008*; *Baselga and Swain, 2009*). We injected 5–7-week-old MMTV-*tva* mice (MA) intraductally with RCAS ($10^8$ IUs per gland) expressing either a constitutively activated version of *Erbb2* (RCAS-*caErbb2*) (*Reddy et al., 2010*) or *Wnt1* (RCAS-*Wnt1*) (*Dunn et al., 2000*). This dosage leads to infection of approximately 0.3% of the luminal epithelial cells (*Du et al., 2006*). 4 to 7 days later, half of the mice were impregnated and then allowed to lactate for 3 weeks. By introducing the oncogene before exposing the mice to a full-term pregnancy, we ensured that an equal number of cells expressed the oncogene in both parous and control virgin mammary glands. This approach allowed us to examine the effects of pregnancy on preexisting precancerous cells and on breast cancer risk while keeping aging-associated variables constant. While caErbB2 induced tumors much more rapidly than Wnt1, each parous group developed tumors significantly faster than its corresponding virgin control cohort (*Figure 1A*; *Figure 2A*) and with higher tumor multiplicity (*Figure 1B*; *Figure 2B,C*). Of note, this experimental design was intended to test the effect of pregnancy on precancerous lesion progression and long-term breast cancer risk, and not on pregnancy-associated breast cancer (PABC). The great majority of tumors initiated by *Erbb2* appeared at least 7 weeks after the completion of pregnancy, beyond the equivalent window of time considered to be PABC in women (*Borges and Schedin, 2012*), and all tumors initiated by *Wnt1* appeared more than 36 weeks after the completion of pregnancy. In accord with their longer latency, the great majority of tumors arising in our mouse models did not display any of the cardinal features of PABC, including aggressive growth rate, high Ki67, and stromal involvement (data not shown) (*Schedin, 2006*). Therefore, we conclude that pregnancy stimulates the long-term cancer risk from mammary cells that have already acquired an oncogenic mutation.

### A full-term pregnancy promotes growth of preexistent premalignant lesions

We next investigated the underlying mechanism of increased breast cancer risk in these parous mice. Tumor growth rates and angiogenesis were comparable between the parous and virgin groups (data not shown), indicating that the impact of pregnancy on tumorigenesis occurs primarily prior to the formation of frank tumors. To identify the effect of pregnancy on premalignant lesions, we compared mammary glands of RCAS-*caErbb2*-infected mice at 1.5, 5.5, and 8.5 weeks post infection, which are equivalent to pregnancy day 7.5 (P7.5), lactation day 12 (L12), and involution day 10 (I10) in the parous group (*Figure 1C*). At 1.5 weeks post injection, the number of total early lesions (defined as any hyperplastic ductal foci comprised of more than three layers of epithelial cells that stained positive for the provirus-encoded oncogene product) was similar between the virgin and P7.5 mice (*Figure 1D*), consistent with their similar rates of initial infection. By 5.5 and 8.5 weeks post-viral injection, however, the total number of early lesions was greater in L12 and I10 groups than in the respective age-matched virgin mice (*Figure 1D*). By 8.5 weeks post infection, the average lesion area was also significantly larger in I10 mice than in the age-matched virgin animals (*Figure 1C,E*). Therefore, we conclude that pregnancy accelerates carcinogenesis by promoting the advancement of early lesions from cells that have already activated oncogenic signaling. We confirmed that these parity-induced effects were not due to incidental increase of RCAS-expressed oncogene (*Figure 1—figure supplement 1*). We also demonstrated that this observation was not caused incidentally by a particular vectoring system: using intraductal injection of a lentiviral vector (*Bu et al., 2009*) to introduce *caErbb2* to ductal epithelia, we also observed accelerated tumorigenesis in the parous group relative to the virgin group (*Figure 1—figure supplement 2*).

We then studied the underlying mechanism of this pregnancy-mediated acceleration of early lesion development. Early lesions in P7.5 and virgin control mice had similar proliferative indices based on staining for Ki67 (*Figure 1—figure supplement 3*) and pHistone3 (data not shown), suggesting that the already high proliferation in oncogene-activated cells cannot be further elevated by pregnancy. Proliferation remained comparable between the early lesions of L12 mice and those of age-matched virgin mice. It was even lower in the early lesions of I10 mice than in those of age-matched virgin mice, suggesting that the molecular network activated in normal epithelia at involution to arrest the cell

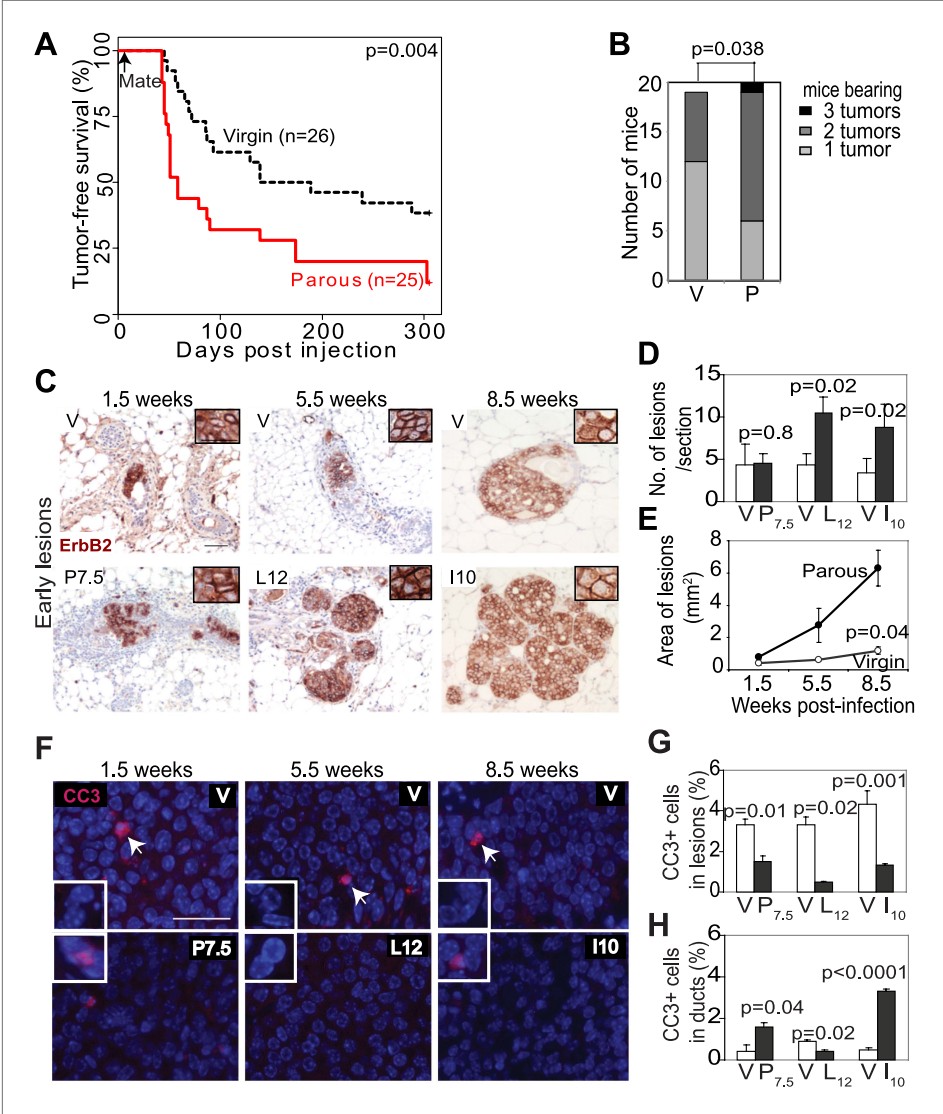

**Figure 1**. Pregnancy promotes survival and carcinogenesis of mammary cells that have already activated ErbB2. (**A** and **B**) Kaplan–Meier tumor-free survival curves (**A**) and tumor multiplicity (**B**). (**C–E**) Identification of early lesions by immunohistochemical staining for the HA tag of RCAS-*caErbb2* (**C**), and quantification of the number (**D**) and area (**E**) of lesions. n ≥ 3 mice for each group. Scale bar = 50 μm. Inset scale bar = 20 μm. (**F–H**) Immunofluorescence for cleaved caspase 3 (CC3, white arrows) in lesions. Insets show CC3 in normal ducts for comparison. Percentages of CC3+ cells in lesions (**G**) and normal ducts (**H**) are shown. Scale bar = 20 μm. n ≥3 mice for each group. The generalized Gehan–Wilcoxon and Fisher's Exact tests generated the p values for (**A**) and (**B**), respectively. Student's *t* test derived all other p values. All columns indicate the mean, and error bars represent SEM except in (**B**). Results from experiments showing similar oncogene expression levels in parous and virgin mice, as well as from comparison of cell proliferation are presented in *Figure 1—figure supplements 1–3*.

The following figure supplements are available for figure 1:

**Figure supplement 1**. Promotion of breast cancer by pregnancy is not caused by increased oncogene expression.

**Figure supplement 2**. Promotion of breast cancer by pregnancy is not caused by the vector system used to induce oncogene expression.

**Figure supplement 3**. Promotion of breast cancer by pregnancy is not caused by increased premalignant cell proliferation.

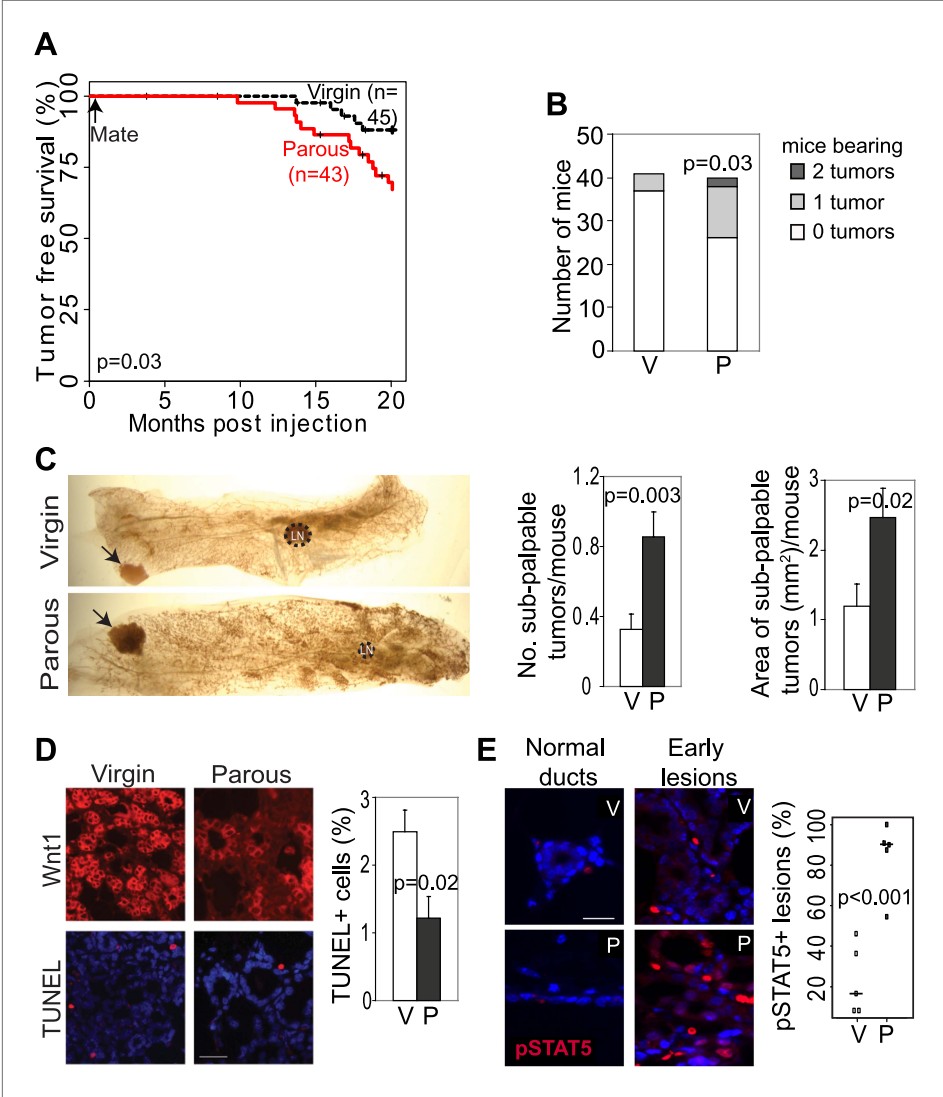

**Figure 2**. Pregnancy promotes development of early lesions and tumors from preexisting mammary cells that overexpress *Wnt1*. (**A** and **B**) Kaplan–Meier survival curves (**A**) and bar graph showing tumor multiplicity (**B**). Generalized Gehan–Wilcoxon test determined p value for (**A**), and Pearson's Chi-square test derived p value for (**B**). (**C**) Area and incidence of sub-palpable tumors (indicated by arrows) were calculated at 18-months post infection using ImageJ. n = 24 virgin and 30 parous mice. Student's *t* test defined p values. LN, lymph node. (**D**) Immunofluorescence for the HA tag (top panel) located the lesions initiated by RCAS-*Wnt1*, and TUNEL assay (bottom panel) performed on the consecutive section identified apoptotic cells in lesions. n = 5 mice. Student's *t* test determined p values. (**E**) Immunofluorescence for pSTAT5. Graph indicates the proportion of pSTAT5+ lesions (>5% pSTAT5+ cells). Horizontal bars represent the mean. n ≥3 mice for each group. Student's *t* test determined p values. Percentage of pSTAT5+ cells in lesions is shown in associated *Figure 2—figure supplement 1B*. For all bar graphs except (**B**), columns represent mean, and error bars represent SEM. All scale bars = 20 μm.

The following figure supplements are available for figure 2:

**Figure supplement 1**. RCAS-*Wnt1*-induced early lesions of parous mice have fewer Ki67+ cells but more pSTAT5+ cells than the lesions of control virgin mice.

**Figure supplement 2**. Early age pregnancy promotes STAT5 activation in precancerous mammary epithelial cells of MMTV-*Erbb2* mice.

cycle (*Figure 1—figure supplement 3B*) also operates in these premalignant cells. Collectively, these results suggest that pregnancy-induced acceleration of carcinogenesis from mutated cells does not result from increased proliferation.

## A full-term pregnancy promotes the survival of preexistent oncogene-activated cells

In response to an oncogenic insult, normal cells usually rapidly activate apoptosis, thus erecting a 'barrier' to carcinogenesis (*Lowe et al., 2004*; *Halazonetis et al., 2008*). We have reported potent apoptosis induction in mammary early lesions of virgin mice initiated by RCAS-*caErB2* (*Reddy et al., 2010*). Therefore, we asked whether this apoptotic reaction was impaired as a result of pregnancy. Early lesions from the above cohorts of RCAS-*caErbb2*-infected mice were stained for cleaved caspase 3 (*Figure 1F*) and also examined with the TUNEL assay (data not shown). As expected, apoptosis was activated in the early lesions of virgin mice at all three time points (1.5, 5.5, and 8.5 weeks post infection) compared to normal ducts (*Figure 1G,H*). However, the level of apoptosis was significantly lower in early lesions of the parous group at all three time points than those in early lesions of the respective virgin mice (*Figure 1G*); it became either comparable to the baseline level in normal ducts during pregnancy and lactation or even significantly lower than that during involution (*Figure 1H*). These observations demonstrate that with a pregnancy preexisting premalignant mammary cells overcome both the apoptotic response to an oncoprotein and the robust cell death at involution, which normally clears the large mass of cells gained during pregnancy/lactation.

## Pregnancy-induced increase of premalignant cell survival is accompanied by STAT5 activation

Apoptosis in oncogene-activated cells is typically initiated by p53 that transcriptionally activates genes encoding apoptosis effectors including Bax, Bak, PUMA and NOXA (*Green and Kroemer, 2009*). However, despite their very low apoptotic rate, the early lesions of I10 mice harbored abundant numbers of cells positive for p53, Bax, and Bak (data not shown), suggesting that as in virgin mice oncogene-activated cells in parous glands are fully capable of activating p53 and its pro-apoptosis targets, but these precancerous cells nevertheless evade apoptosis.

We next asked whether evasion of apoptosis by these precancerous cells specifically in the parous group is due to activation of prosurvival machinery that antagonizes the pro-apoptosis targets of p53. The STAT5 (signal transducer and activator of transcription) transcriptional factor plays a crucial role in normal mammary cell survival as well as proliferation and differentiation (*Liu et al., 1997*; *Iavnilovitch et al., 2002*). While found in some mammary cells prior to pregnancy, the activated form of STAT5 is detected in the great majority of epithelial mammary cells, and at increased levels, during pregnancy and lactation (*Liu et al., 1995*; *Wagner and Rui, 2008*). This is because placental lactogen and prolactin (PRL) activate PRLR, leading to the recruitment of Jak2 and Jak2-mediated phosphorylation and activation of STAT5 (*Hennighausen and Robinson, 1998*; *Haricharan and Li, 2014*). At the onset of involution, STAT5 is rapidly deactivated to allow programmed death of excess cells accumulated during pregnancy and lactation (*Li et al., 1997*; *Hennighausen and Robinson, 2001*; *Kreuzaler et al., 2011*). Transgenic expression of constitutively active *Stat5* mutants in mammary glands leads to increased mammary cell survival, hyperplasia, and even occasional tumors (*Iavnilovitch et al., 2002*; *Dong et al., 2010*; *Vafaizadeh et al., 2012*), while *Stat5* heterozygosity delays mammary tumorigenesis in mice carrying a transgenic oncogene (*Humphreys and Hennighausen, 1999*; *Ren et al., 2002*). pSTAT5 is also detected frequently in human breast early lesions (ductal carcinomas in situ) and to a lesser degree in invasive breast cancers (*Cotarla et al., 2004*; *Nevalainen et al., 2004*; *Peck et al., 2011*).

Therefore, we asked whether preexisting precancerous cells in our parous cohort might evade apoptosis by exploiting this normally well-regulated cell survival pathway. The average percentage of pSTAT5[+] cells per lesion was significantly higher in the parous mice at P7.5, L12, and I10 than in the corresponding age-matched virgin groups, while cells positive for total STAT5 were readily detectable with comparable frequency in virgin vs parous lesions as expected (*Kazansky et al., 1995*) (*Figure 3A*, *Figure 3—figure supplement 1*). Furthermore, >70% of the preneoplastic lesions in the parous cohort harbored >5% pSTAT5[+] cells (*Figure 3B*) and were thus defined as pSTAT5[+] lesions (this cut-off was chosen because pSTAT5 positivity at this frequency correlated significantly with increased survival as shown below in *Figure 3C*), while <25% of the lesions in the virgin cohort were scored pSTAT5[+]. These

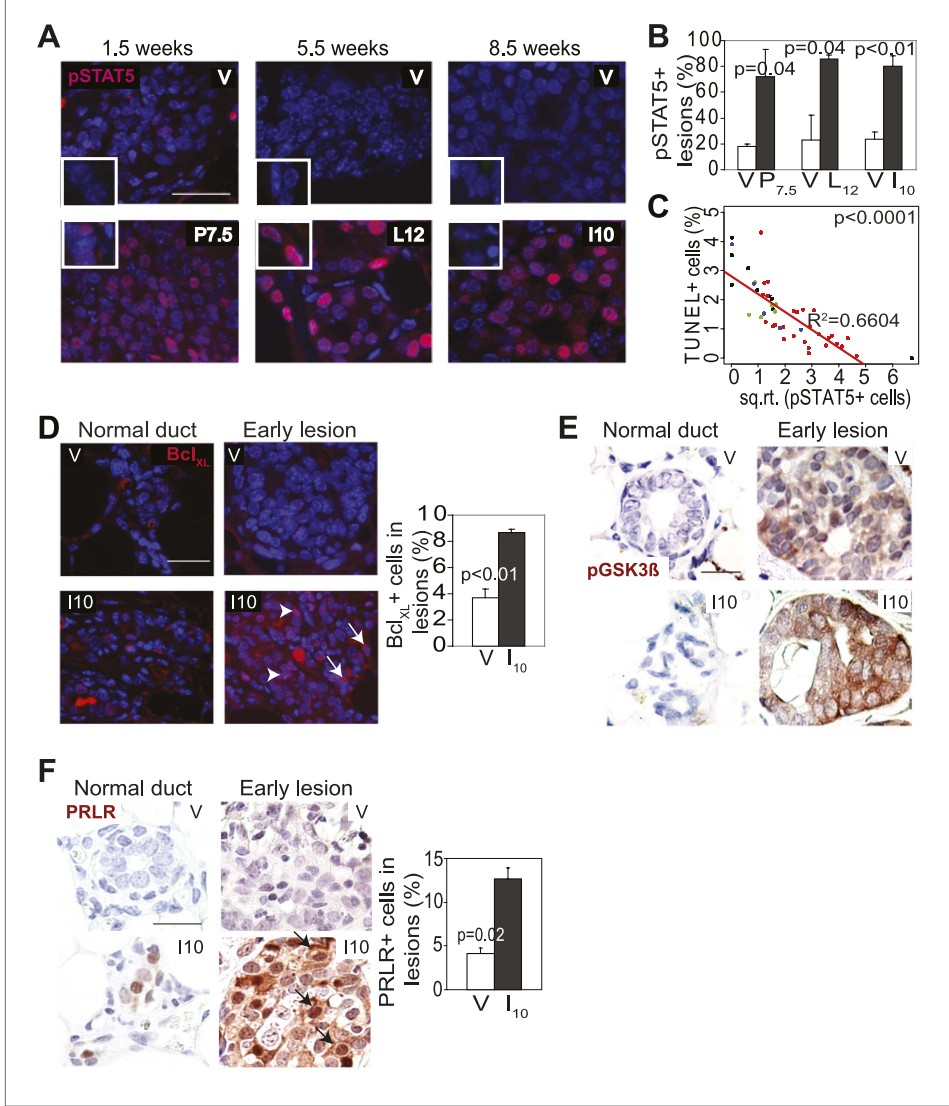

**Figure 3**. Pregnancy causes preexisting early lesions to persistently activate STAT5 signaling. (**A** and **B**) Immunofluorescence for pSTAT5 in lesions (**A**) and accompanying quantification (**B**). Insets show staining of normal ducts (**A**). pSTAT5+ lesions are those that have >5% pSTAT5+ cells (**B**). n ≥ 3 mice. Quantification of percentage of pSTAT5+ cells in lesions is presented in associated *Figure 3—figure supplement 1* and analysis of pSTAT1, pSTAT3 and pSTAT6 is presented in *Figure 3—figure supplement 3*. (**C**) Generalized linear regression analysis for correlation between the square root of the number of pSTAT5+ cells in individual I10 lesions and percentage of CC3+ cells in their corresponding lesions. Dots represent individual lesions. Colors represent individual mice (n = 4). (**D**) Immunofluorescence for BclXL. Positive staining in cytoplasmic (arrow) and perinuclear (arrowhead) regions was observed. Bar graph shows the percentage of BclXL+ cells in early lesions. n = 4. Supporting data in *Figure 3—figure supplement 2*. (**E**) Immunohistochemical staining for GSK3-β phosphorylated at Ser9. n = 4 mice. (**F**) Immunohistochemical staining for PRLR. Black arrows indicate cells with membrane PRLR. Bar graph shows the percentage of PRLR+ cells in early lesions. n = 4 mice. Supporting data in *Figure 3—figure supplement 4*. Scale bars = 20 μm. All columns indicate the mean, and error bars represent SEM. p value for the P7.5 time point was derived by a Wilcoxon Rank Sum test. All other p values were generated by Student's *t* test. Supporting Western blotting data are presented in associated figure supplements.

The following figure supplements are available for figure 3:

**Figure supplement 1**. *caErbb2*-induced early lesions activate STAT5 more robustly in parous mice than in virgin mice.

*Figure 3. Continued on next page*

*Figure 3. Continued*

**Figure supplement 2**. *caErbb2*-induced early lesions activate downstream pSTAT5 targets more robustly in parous mice than in virgin mice.

**Figure supplement 3**. STAT1, STAT3 and STAT6 activation is minimal and comparable between *cErbb2*-induced lesions of virgin and parous mice.

**Figure supplement 4**. *caErbb2*-induced early lesions activate PRLR persistently in parous mice.

data suggest that premalignant mammary cells respond to pregnancy and lactation hormones by activating STAT5, as their normal counterparts do (*Figure 3A* inset); however—unlike their normal counterparts—they continue to maintain high levels of pSTAT5 even during involution (*Figure 3—figure supplement 1B*).

Consistent with STAT5 activation, when early lesions in I10 mice were further analyzed, we detected increased levels of the STAT5 prosurvival transcriptional targets $Bcl_{XL}$ (*Socolovsky et al., 2001*) and $Bcl_2$ (*Lord et al., 2000*; *Hand et al., 2010*) as well as cyclinD1 (*Matsumura et al., 1999*; *Brockman et al., 2002*) compared to early lesions in age-matched virgin mice (*Figure 3D*, *Figure 3—figure supplement 2*). Importantly, there was a strong inverse correlation between the percentage of pSTAT5$^+$ cells and the level of apoptosis in these parous early lesions (*Figure 3C*), suggesting a causal relationship between pSTAT5 levels and cell survival. Of note, <1% of cells in virgin and parous early lesions were pSTAT1$^+$ or pSTAT6$^+$, and <2% were pSTAT3$^+$ (*Figure 3—figure supplement 3*), indicating that among the STATs known to be activated in the mammary gland (*Chapman et al., 1999*; *Watson, 2001*; *Hennighausen and Robinson, 2005*; *Khaled et al., 2007*), STAT5 is the predominant family member activated in parous early lesions.

We also tested whether increased cell survival and pSTAT5 occur in parous early lesions initiated by a different oncogene. In virgin mice infected by RCAS-*Wnt1*, early lesions were not readily identified on an H&E-stained section within the first few months of infection, but by 6 months post infection, early lesions were easily detected. Compared to the lesions in virgin mice, the Wnt1-initiated lesions in age-matched parous mice exhibited improved survival (*Figure 2D*) and an increased percentage of pSTAT5$^+$ cells (*Figure 2E*, *Figure 2—figure supplement 1B*), while also showing reduced proliferation (*Figure 2—figure supplement 1A*) similar to their RCAS-*caErbb2*-injected counterparts.

In addition, we asked whether parity leads to persistent STAT5 activation in classical transgenic models of breast cancer. pSTAT5 was readily detected in mammary epithelia of MMTV-*Erbb2* transgenic mice at involution day 10, but not in mammary glands of age-matched non-transgenic virgin mice (*Figure 2—figure supplement 2*). Together, these data suggest that parity-induced persistent activation of STAT5 in precancerous cells is not limited to a specific oncogenic mutation or a unique model of breast cancer.

We next tested whether persistent STAT5 signaling in premalignant mammary cells is due to aberrant activation of upstream components that normally regulate STAT5 activity during pregnancy and lactation. As shown in *Figure 3F*, while being diminished in normal epithelia by I10, PRLR was readily detected in *caErbb2*-induced early lesions, with the percentage of positive cells being threefold higher than that in early lesions of age-matched virgin mice (*Figure 3F*, *Figure 3—figure supplement 4B*). This increase was confirmed by Western blotting comparing protein extracts of early lesions-bearing whole mammary glands of parous and virgin mice (*Figure 3—figure supplement 4A*). These data suggest that oncogene-activated mammary cells, unlike their normal counterparts, fail to degrade PRLR at the onset of involution.

PRLR is normally deactivated by GSK3β-initiated phosphorylation and proteosomal degradation as circulating prolactin diminishes at involution (*Plotnikov et al., 2009*). However, transformed cells frequently phosphorylate and inactivate GSK3β and thus can aberrantly maintain PRLR (*Plotnikov et al., 2008*). Therefore, we next tested whether oncogenic signaling inactivated GSK3β in early lesions, thereby allowing the PRLR molecules activated specifically in the parous mice to remain at the membrane longer than otherwise expected. Indeed, the inactivated form of GSK3β, pS9-GSK3β, was readily detected by immunohistochemistry in *caErbb2*-initiated early lesions in parous mice while it was not found in normal mammary cells (*Figure 3E*). In early lesions of virgin mice, an elevated and

similar level of pS9-GSK3β was also detected due to ErbB2-mediated oncogenic signaling. Together, these data suggest that during pregnancy and lactation preexisting premalignant mammary cells activate the PRLR-Jak2-STAT5 signaling cascade as normal mammary cells do, but after the onset of involution, these oncogene-activated cells aberrantly maintain an activated state of this pathway likely via oncoprotein-mediated inactivation of GSK3β preventing timely PRLR degradation. However, precancerous cells in virgin mice do not activate PRLR and therefore cannot exploit ErbB2-mediated GSK3β inactivation for aberrant activation of PRLR-STAT5. Of note, by 6 months post involution, as few pSTAT5+ cells (5.7 ± 2.2%) could be detected in early lesions of parous mice, as in virgin mice (p=0.3; n = 4 mice; data not shown), suggesting that with time, parity-induced elevated levels of pSTAT5 in parous mice subside.

## STAT5 activation is sufficient to recapitulate pregnancy's promotion of breast cancer

To directly test whether STAT5 activation is sufficient to mimic pregnancy in promoting the survival and tumorigenesis of oncogene-activated mammary cells, we used three complementary in vivo approaches. First, we asked whether forced activation of STAT5 in virgin mice lowers apoptotic levels in caErbB2-expressing mammary epithelial cells. We used RCAS-*caStat5a* (*Dong et al., 2010*) and RCAS-*caErbb2* viruses to co-infect MMTV-*tva* virgin mice (the co-infection efficiency is 8 ± 4%; data not shown). At 2 weeks post infection, the resulting early lesions displayed either caErbB2 alone (69%) or both caErbB2 and caSTAT5a (31%), and none had caSTAT5a alone (data not shown), confirming our previous observation that the activation of STAT5 alone is not strongly tumorigenic (*Dong et al., 2010*). Enrichment of early lesions positive for both caErbB2 and caSTAT5a relative to the low frequency of co-infection suggests that caSTAT5a promotes lesion initiation by caErbB2. Importantly, the lesions that were positive for both caErbB2 and caSTAT5a were less apoptotic than the lesions induced in the contralateral glands by caErbB2 alone (*Figure 4—figure supplement 1A*). Proliferation rates in these two sets of lesions were similar (*Figure 4—figure supplement 1B*). As expected, the mice that were infected only with RCAS-*caStat5a* did not tumors, while the co-infected mice developed tumors much more rapidly than the mice infected with RCAS-*caErbb2* alone (p=0.0002; *Figure 4A*), consistent with previous reports of a pro-tumorigenic role for STAT5 in the breast (*Furth et al., 2011*). Significantly, the Kaplan–Meier tumor-free survival curve of the co-infected virgin mice became superimposable on that of the parous cohort of mice from *Figure 1A* that was infected with RCAS-*caErbb2* alone and subsequently impregnated (*Figure 4A*, red line). These results suggest that forced STAT5 activation in virgin mice is sufficient to mimic breast cancer promotion by pregnancy.

We then asked whether physiological levels of STAT5 activity are sufficient to mimic pregnancy's impact on mammary tumorigenesis initiated by caErbB2. Among the experimental MMTV-*tva* virgin mice that were euthanized at 8.5 weeks post RCAS-*caErbb2* infection (described in *Figure 1C–E*), a small subset (20%) exhibited high baseline levels of pSTAT5 in both the normal mammary epithelium and early lesions, while the rest had barely detectable pSTAT5 in either (*Figure 4—figure supplement 2A*). We termed these mice the pSTAT5hi and pSTAT5lo groups, respectively. Apoptosis levels were inversely correlated with levels of pSTAT5 in early lesions from these two groups of mice (*Figure 4—figure supplement 2B*). Moreover, when we reanalyzed the data among the experimental virgin mice from *Figure 1A*, the pSTAT5hi sub-cohort exhibited a shorter tumor-free survival than the pSTAT5lo mice and, remarkably, became indistinct from that of the parous cohort from *Figure 1A* (*Figure 4B*). These observations suggest that high baseline pSTAT5 levels in virgin mice mimic stimulation of breast carcinogenesis by pregnancy. Not surprisingly, in the parous group from *Figure 1A*, mice with pSTAT5lo and pSTAT5hi baseline levels (same criteria as in virgin mice) had similar tumor latency (data not shown), probably because profound STAT5 activation during pregnancy and lactation masks the effect of baseline pSTAT5 variations among individuals.

To directly demonstrate that in virgin mice high levels of endogenous pSTAT5 can mimic pregnancy in increasing breast cancer risk, we took advantage of our newly characterized transgenic WAP-*tva* mouse line (*Haricharan et al., 2013*), which expresses *tva* from the promoter of the *Wap* (whey acidic protein) gene—a classical transcriptional target of STAT5 (*Hennighausen and Robinson, 2008*)—and used it for selective delivery of *caErbb2* into the pSTAT5+ subset of mammary epithelial cells in virgin mice. As predicted, the infected cells in this line harbored significantly more pSTAT5+ cells (threefold) than did the infected cells in MMTV-*tva* mice, while the pSTAT5+ proportion among the uninfected cells remained comparable between the two lines of mice (*Figure 4—source data 1*). We

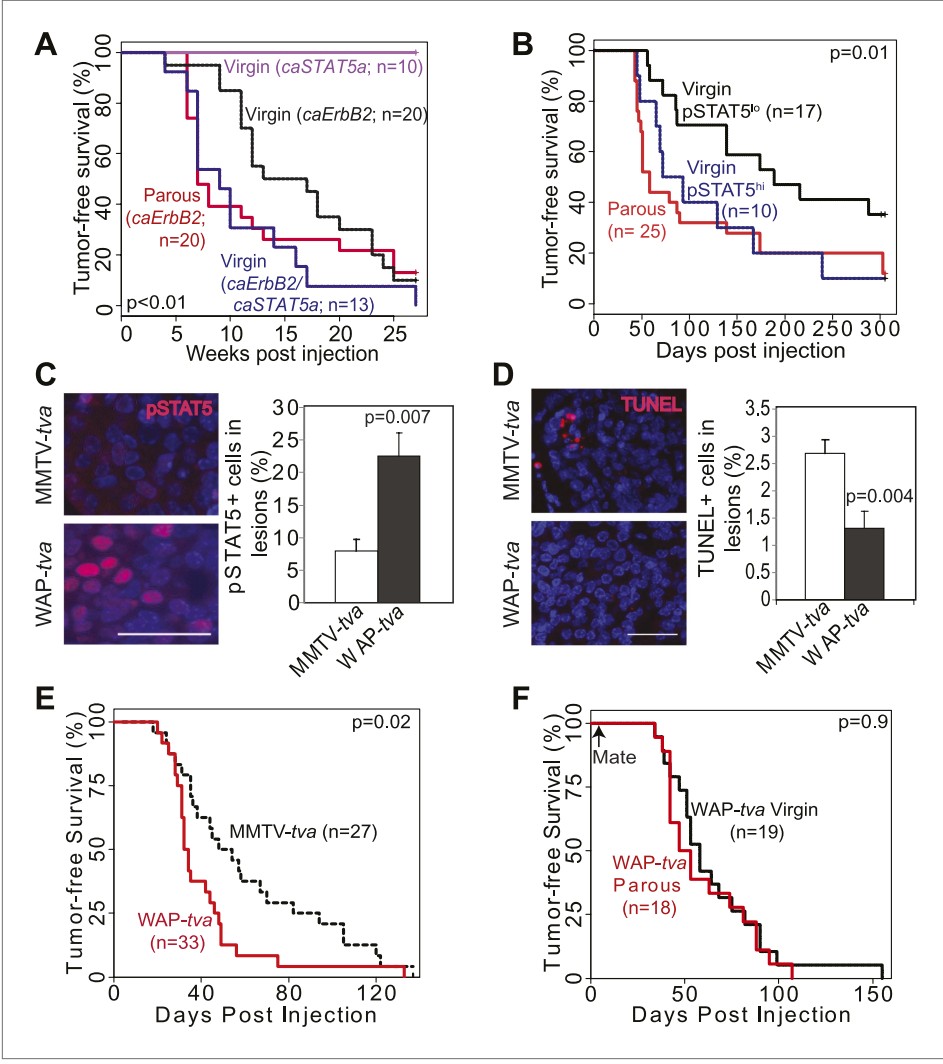

**Figure 4**. STAT5 activation in virgin mice mimics pregnancy's promotion of *caErbb2*-induced mammary tumorigenesis. (**A**) Kaplan–Meier tumor-free survival curves of virgin mice injected with RCAS-*caErbb2* alone (black) or with both RCAS-*caErbb2* and RCAS-*caStat5a* (blue). The parous group from *Figure 1A* is shown for comparison (red). Comparison of lesion apoptosis and proliferation between the two virgin groups is presented in associated *Figure 4—figure supplement 1*. (**B**) Kaplan–Meier tumor-free survival curves of the virgin cohort from *Figure 1A* that was stratified into pSTAT5hi (blue) and pSTAT5lo (black) groups based on baseline pSTAT5 levels in the normal mammary glands of each of these mice (*Figure 4—figure supplement 2*). The parous group from *Figure 1A* is shown for comparison (red). (**C** and **D**) Immunofluorescence detected pSTAT5+ cells (**C**) and apoptotic cells (**D**) in lesions induced by RCAS-*caErbb2* in MMTV-*tva* and WAP-*tva* mice. n = 6 (**C**) and 4 (**D**) mice. Scale bar=20 μm. Columns represent the mean, and error bars indicate SEM. Student's *t* test determined p values. Proliferation in lesions is shown in *Figure 4—figure supplement 3*, and relevant baseline characteristics of these two mouse lines are delineated in *Figure 4—source data 1*. (**E**) Kaplan–Meier tumor-free survival curves comparing MMTV-*tva* and WAP-*tva* virgin mice injected with RCAS-*caErbb2*. Tumor multiplicity is presented in *Figure 4—figure supplement 4*. (**F**) Kaplan-Meier tumor-free survival curves comparing WAP-*tva* mice injected with RCAS-*caErbB2* and then either kept as virgin or impregnated. The p value for all tumor-free survival comparisons was generated using generalized Gehan-Wilcoxon test.

The following source data and figure supplements are available for figure 4:

**Source data 1**. Comparison of uninfected and infected cells in MMTV-*tva* vs WAP-*tva* mice.

**Figure supplement 1**. Exogenous STAT5 activation in virgin mice recapitulates pregnancy's promotion of breast cancer.

*Figure 4. Continued on next page*

*Figure 4. Continued*

**Figure supplement 2**. pSTAT5 in normal ducts is correlated with pSTAT5 in early lesions; CC3 in early lesions is inversely correlated with pSTAT5.

**Figure supplement 3**. caErbB2 leads to early lesions with similar levels of proliferation in WAP-tva virgin mice and MMTV-tva virgin mice.

**Figure supplement 4**. caErbB2 leads to a higher tumor multiplicity in WAP-*tva* virgin mice than in MMTV-*tva* virgin mice.

further confirmed that RCAS infected equivalent numbers of mammary epithelial cells in WAP-*tva* and MMTV-*tva* mice (*Figure 4—source data 1*). As predicted, there were many more pSTAT5⁺ cells in the resulting early lesions in WAP-*tva* mice than in MMTV-*tva* mice (*Figure 4C*). The apoptosis rate was lower in the early lesions in WAP-*tva* mice (*Figure 4D*), while proliferation remained comparable (*Figure 4—figure supplement 3*). Mammary tumors also arose more rapidly (*Figure 4E*) and with greater multiplicity (*Figure 4—figure supplement 4*) in WAP-*tva* mice than in MMTV-*tva* mice, consistent with previous evidence of alveolar or alveolar progenitor cells as the preferred cell of origin for ErbB2-initiated mammary tumors (*Li et al., 2003*; *Andrechek et al., 2004*; *Henry et al., 2004*; *Jeselsohn et al., 2010*). These data demonstrate that the STAT5-activated subset of mammary epithelial cells has a compromised apoptotic response to oncogenic insult and is highly vulnerable to oncogene-induced carcinogenesis even without a pregnancy. More importantly, in WAP-*tva* mice infected by RCAS-*caErbb2*, pregnancy could no longer accelerate the already hastened tumorigenesis (*Figure 4F*), directly implicating STAT5 in mediating pregnancy's stimulation of breast carcinogenesis. Together, we conclude that STAT5 activation during pregnancy is sufficient to increase the survival of precancerous cells and to promote their progression to cancer.

## STAT5a is mandatory for pregnancy's promotion of tumorigenesis from preexisting precancerous cells

To further test the role of STAT5 in mediating pregnancy's stimulation of carcinogenesis from preexisting precancerous cells, we asked whether the gene encoding STAT5a, the predominant form of STAT5 in the mammary gland and in the lesions generated in our model system (*Figure 3—figure supplement 1*), is required for the observed impact of pregnancy on early lesions. The *Stat5a* knockout mice on the FVB background showed normal mammary development during pregnancy, lactation, and involution (*Figure 5—figure supplement 1*) unlike those on the 129 background (*Liu et al., 1997*). When these *Stat5a⁻/⁻* mice were bred to MMTV-*tva* mice, infected with RCAS-*caErbb2*, impregnated 4 days post infection, and analyzed at I10, the resultant early lesions exhibited significantly lower levels of pSTAT5 than did the lesions in *Stat5a⁺/⁺* mice (*Figure 5A*) confirming the predominance of STAT5a in the mammary gland. These lesions showed a level of apoptosis both comparable to that of early lesions in the age-matched virgin control mice and significantly higher than that of early lesions of *Stat5a⁺/⁺* mice at I10 (*Figure 5B*). The number and size of early lesions were also similar in parous vs virgin *Stat5a⁻/⁻* mice (*Figure 5C,D*). Importantly, tumor incidence and tumor latency were similar in parous vs virgin *Stat5a⁻/⁻* mice (*Figure 5E,F*). Together, these data demonstrate that *Stat5a* is necessary for pregnancy-mediated promotion of preneoplastic cell survival, lesion progression, and tumorigenesis. Of note, we did not observe a difference in frequency of pSTAT1⁺, pSTAT3⁺, or pSTAT6⁺ cells in lesions of parous *Stat5a⁻/⁻* animals when compared to lesions of wild-type animals (p=0.4, 0.5, and 0.3, respectively; *Figure 5—figure supplement 2*).

## Short-term treatment of lactation-completed mice with inhibitors of STAT5 signaling prevents pregnancy's promotion of breast tumorigenesis

To build preclinical evidence for clinical trials to test inhibitors of STAT5 signaling for lowering breast cancer risk in women who have undergone a late-age pregnancy or have abnormally high baseline levels of pSTAT5 signaling, we tested the efficacy of pharmacological inhibitors in preventing mammary tumorigenesis in our mouse model. We first used two inhibitors to block the upstream activators of STAT5 and assessed the effect of these inhibitors on lesion cell apoptosis and lesion progression. AG490 is a small molecule inhibitor that blocks several tyrosine kinases including Jak2 (*Gazit et al.,*

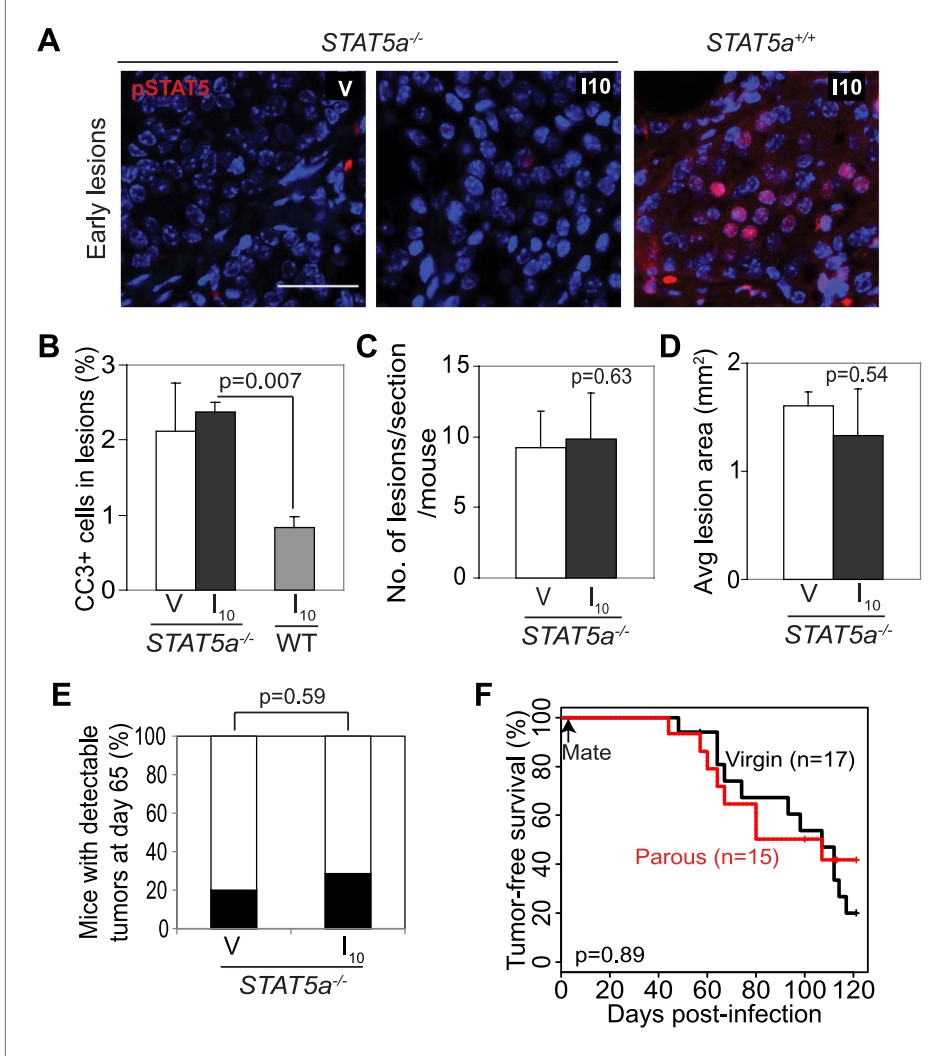

**Figure 5**. *Stat5a* genetic ablation negates pregnancy's promotion of caErbB2-initiated mammary tumorigenesis. (**A**) Immunofluorescent staining for pSTAT5 in early lesions of mice with the indicated genotype. n = 5. Scale bar = 20 μm. (**B**) Levels of apoptosis in early lesions were quantified via immunostaining for CC3. n = 5 mice. (**C** and **D**) Average lesion number (**C**) and area (**D**) were quantified using images of immunostaining for RCAS-*caErbb2*-HA. n = 5 mice. (**E**) Tumor incidence at day 66 post injection with RCAS-*caErbb2*. Black region indicates percentage of mice with palpable tumors. n = 18 parous mice and 19 virgin mice. (**F**) Kaplan–Meier tumor-free survival curves comparing *Stat5a*−/− parous and virgin mice. p values were determined by Student's *t* test (**B–D**), Fisher's exact test (**E**), and generalized Gehan–Wilcoxon test (**F**). Columns represent means and error bars SEM except for (**E**).
The following figure supplements are available for figure 5:

**Figure supplement 1**. *Stat5a* ablation (FVB background) does not affect mammary gland development during pregnancy, lactation, and involution.

**Figure supplement 2**. *Stat5a* ablation (FVB background) does not affect pSTAT1, pSTAT3, or pSTAT6 positivity in early lesions of parous mice at involution day 10.

*1991*); Ruxolitinib is a small molecule inhibitor that specifically suppresses Jak kinases (**Quintas-Cardama et al., 2010**). AG490 (75, or 200 mg/kg body weight) (**Levitzki and Gazit, 1995**) or diluent (DMSO) was administered every other day for 5 days starting from I2 to early lesion-bearing mice that had already completed lactation. AG490 decreased pSTAT5, Bcl$_{XL}$, and cell survival in parous early lesions, and induced lesion regression (**Figure 6A–C**, **Figure 6—figure supplement 1A**). In virgin mice, AG490 did

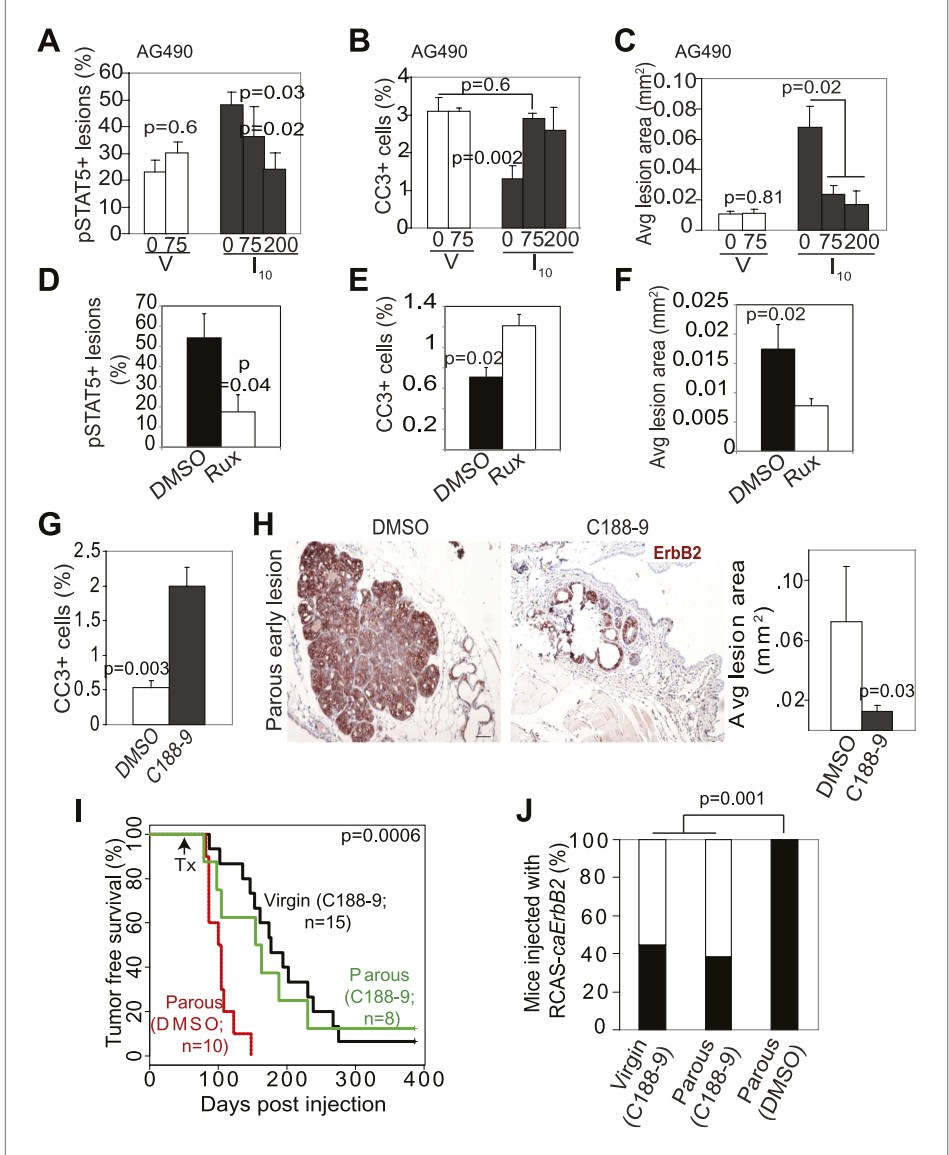

**Figure 6**. Inhibition of STAT5 signaling prevents pregnancy's promotion of breast cancer risk. (**A–F**) Bar graphs showing the percentage of pSTAT5+ RCAS-*caErbb2*-induced lesions (**A** and **D**), apoptotic cells in lesions (**B** and **E**), and average lesion area (**C** and **F**) of I10 or virgin mice that were treated with either AG490 (**A–C**) at the indicated doses or Ruxolitinib (**D–F**). n ≥ 3 mice. Statistical significance was determined by Student's *t* test (**A** and **B**, **D–F**) and ANOVA (**C**). The impact of AG490 on $Bcl_{XL}$ in these lesions is presented in *Figure 6—figure supplement 1*. (**G**) Bar graphs showing percentage of CC3+ cells in lesions of mice treated with DMSO or C188-9. n = 4 mice. Student's *t* test derived p value. The potency of C188-9 on STAT5 is shown in *Figure 6—figure supplement 2*. The impact of C188-9 on $Bcl_{XL}$ and cell proliferation is shown in *Figure 6—figure supplement 3*. (**H**) Early lesions in parous mice treated either with DMSO or C188-9 were identified by immunohistochemical staining for the HA tag of RCAS-*caErbb2*, and their areas were then quantified using ImageJ. n = 4 mice. Scale bar = 50 μm. Student's *t* test derived p value. (**I** and **J**) Kaplan–Meier tumor-free survival curves (**I**) and tumor incidence plot (**J**) of parous and age-matched virgin controls treated with either C188-9 or DMSO. Generalized Gehan–Wilcoxon test determined p-value for (**I**), and Fisher's Exact test determined p-value for (**J**). Tumor latency and incidence for virgin mice treated with C188-9 or DMSO are presented in *Figure 6—figure supplement 4*. For all bar graphs, columns represent the mean, and error bars indicate SEM.

The following figure supplements are available for figure 6:

**Figure supplement 1**. Impact of short-term treatment with AG490 on STAT proteins, $Bcl_{XL}$, and apoptosis in MMTV–tva mice and WAP–tva mice.

*Figure 6. Continued on next page*

*Figure 6. Continued*

**Figure supplement 2**. C188-9, a small molecule inhibitor of STAT signaling, inhibits STAT5 phosphorylation.
**Figure supplement 3**. Impact of short-term C188-9 on biomarkers and lesion multiplicity in parous mice.
**Figure supplement 4**. Short-term C188-9 treatment does not affect tumorigenesis in virgin mice.

not have any significant impact on precancerous cell pSTAT5 positivity, apoptosis or early lesion regression (*Figure 6A–C*), in accordance with the low levels of pSTAT5 in these lesions. This finding also indicates that AG490, which has been reported to inhibit EGFR and ErbB2, did not inhibit oncogenic signaling from caErbB2 in our model, perhaps due to the mutations in this ErbB2 variant (*Bargmann and Weinberg, 1988*). Of note, AG490 did not affect pSTAT1, pSTAT3 or pSTAT6 positivity in either parous or virgin early lesions (*Figure 6—figure supplement 1B*). Using a similar experimental approach, we found that in parous mice ruxolitinib also decreased pSTAT5, blocked cell survival, and induced regression of early lesions (*Figure 6E–F*). Together, these findings suggest that Jak2 signaling is required for early lesion survival and progression and that blocking the upstream tyrosine kinase activity of STAT5 can prevent the progression of early lesions in parous mice.

To directly test the impact of blocking STAT5 activity on early lesion progression in these post-lactational animals, we evaluated the effect of our recently developed direct, small molecule STAT inhibitor (C188-9) (*Xu et al., 2009*; *Redell et al., 2011*). This inhibitor targets the phosphotyrosyl peptide-binding pocket within the SH2 domain of STAT5, STAT3, and possibly other STAT proteins, thereby blocking two steps in their activation—recruitment to ligand-activated receptor complexes and dimerization (*Figure 6—figure supplement 2*). C188-9 (100 mg/kg BW) or diluent (DMSO) was injected intraperitoneally into mice bearing RCAS-*caErbb2*-initiated early lesions only twice (I2 and I9). 1 day after the second injection (I10), C188-9 caused a significant decrease in pSTAT5$^+$ and Bcl$_{XL}^+$ cells (*Figure 6—figure supplement 3A*) but did not significantly reduce the low frequency of cells positive for pSTAT1, pSTAT3, or pSTAT6 (p=0.2, 0.6, and 0.8, respectively; *Figure 6—figure supplement 3B*). Further, C188-9 caused a significant rise in apoptosis in the lesions of parous mice, (*Figure 6G*) while having no detectable effect on proliferation (*Figure 6—figure supplement 3C*). Importantly, C188-9 caused a dramatic regression of parous premalignant lesions (*Figure 6H*, *Figure 6—figure supplement 3D*). This effect is unlikely to be caused by off-target impact: in virgin mice, even four doses of C188-9 did not result in a significant decrease of pSTAT5$^+$ cells, increase of apoptosis, or decrease of size in early lesions (p=0.6, 0.8, and 0.7, respectively; n = 3 mice; data not shown). Therefore, C188-9, like AG490, can block mammary early lesion survival and progression specifically in parous mice.

To demonstrate that these damaged early lesions were impaired in their ability to progress to cancer, we generated another cohort of these early lesion-bearing parous mice, and gave them four weekly injections of C188-9 or diluent (DMSO) starting at I2. This short-term treatment led to a significant reduction in tumor incidence: by 3 months post ErbB2 activation, only three of the eight C188-9-treated mice had developed palpable tumors, while all 10 diluent-treated parous mice had done so (*Figure 6J*). For comparison, eight of the 15 C188-9-treated virgin control mice had also developed palpable tumors at this time point. This regimen also led to a significant improvement in tumor-free survival in the parous group (*Figure 6I*). In fact, the tumor-free survival curve became superimposable on that of the age-matched virgin group that was similarly treated. It is important to note that C188-9 did not significantly affect either tumor incidence or latency in virgin mice (p=0.6 and 0.5, respectively; *Figure 6—figure supplement 4*). Therefore, a short-term treatment with C188-9 hindered the progression of preexisting early lesions and removed the stimulatory effect of pregnancy on breast cancer risk. Together, these preclinical data suggest that a short-term prophylactic treatment with inhibitors blocking Jak2-STAT5 signaling may lower breast cancer risk for women who have had a late-age first pregnancy and have already completed lactation.

## STAT5 activation during human breast cancer formation is affected by parity and may be a risk factor

The extensive mouse model data presented here suggest that in women a pregnancy may also cause preexisting early lesions to aberrantly and persistently activate STAT5, and chemoprevention targeting

STAT5 activity may lower breast cancer risk in women who have had a late-age pregnancy as well as in those who have abnormally high levels of pSTAT5. Indeed, when we compared 14 cases of DCIS (pre-invasive lesions) from women who had a pregnancy 6–25 years prior to diagnosis with 13 cases from age-matched nulliparous women in a tissue bank from the University of Colorado (*Figure 7—source data 1A*), we detected significantly more pSTAT5+ lesions in the parous cases than in the nulliparous cases (p=0.01; *Figure 7A,B*). Of note, only the cases with >5 years between last pregnancy and diagnosis were included for comparison in order to focus on pregnancy's long-term impact on cancer risk rather than on the relatively rare pregnancy-associated breast cancer subset. Interestingly, patients with higher percentages of pSTAT5+ cells within their lesions also had higher percentages of pSTAT5+ cells in the lesion-adjacent, histopathologically benign breast epithelium (which may be abnormal at genetic, epigenetic and gene expression levels) (*Figure 7C,D*). This observation suggests that pSTAT5 levels in histopathologically 'normal' breast epithelia may predict breast cancer risk in women. Therefore, we also quantified pSTAT5+ cell frequencies in tumor-adjacent breast epithelia (≥5.0 cm away from an invasive cancer) from a cohort of 24 breast cancer patients diagnosed at the M.D. Anderson Cancer Center (*Figure 7E*). Higher pSTAT5 in these breast epithelia retrospectively predicted accelerated development of breast cancer following a full-term pregnancy (p=0.007; *Figure 7F*) with all other characteristics being comparable (*Figure 7—source data 1B*). Although the sample size of these two cohorts is relatively small, they nevertheless suggest an involvement of STAT5 in pregnancy's promotion of long-term risk of sporadic human breast. Of note, BRCA mutation tests were negative in all 10 patients screened in the University of Colorado cohort (27 cases), and in 22 out of 24 patients screened in the MD Anderson Cancer Center cohort.

## Pregnancy causes normal mammary cells to become less vulnerable to transformation by a future oncogenic event

Our mouse model data are not in conflict with the epidemiological observation that a first pregnancy before age 22 greatly reduces breast cancer risk (*MacMahon et al., 1970*), because at this young age, the chance of having already accumulated precancerous cells is small. To ascertain that the normal mammary epithelium in fully involuted mammary glands responds differently to a future oncogenic insult compared to preexistent oncogene-activated cells responding to a pregnancy, we injected RCAS-*caErbb2* into fully involuted mammary glands of MMTV-*tva* mice (age = 20 weeks). The normal mammary cells in these involuted older mice had the same baseline levels of total STAT5a protein and pSTAT5 as the normal mammary cells in the 6-week-old mice used for oncogene introduction prior to a pregnancy (*Figure 8—figure supplement 1*). However, when challenged by an oncogene, the normal mammary epithelial cells in these involuted glands did not aberrantly activate STAT5 (*Figure 8A*) when compared to mammary cells that gained caErbB2 first and were later exposed to pregnancy hormones (*Figure 3A*). Consequently, their pSTAT5 level was comparable to the low level in early lesions arising in age-matched virgin mice (*Figure 8A*). The apoptotic index was also high and comparable to that of the lesions in the age-matched virgin control (*Figure 8B*). Therefore, normal breast epithelial cells in the fully involuted mammary gland—in the event of suffering an oncogenic mutation—do not aberrantly activate STAT5 signaling and can successfully initiate an apoptosis response to evade carcinogenesis.

Furthermore, we found that the mammary cells in these involuted glands responded to caErbB2 with a lower proliferation rate than the cells in virgin mice (*Figure 8C–E*), demonstrating that the slowly proliferating mammary cells of involuted mammary glands are also indolent to aberrant proliferation initiated by a future oncogenic mutation, as previously suggested (*Guzman et al., 1999*; *Medina et al., 2001*).

## Discussion

The results of this study demonstrate that with a pregnancy, preexisting precancerous cells in the breast subvert a normally tightly-controlled STAT5 signaling pathway to benefit their survival and consequently progress more rapidly to cancer (*Figure 8F*). In contrast, if mammary cells activate oncogenic signaling following the completion of a pregnancy and lactation, they acquire no survival advantage over cells that activate oncogenic signaling in nulliparous mice (*Figure 8B*). Moreover, these cells become indolent to oncogene-induced aberrant proliferation (*Figure 8E*). These results, when combined with our data on pregnancy's instigation of preexisting precancerous cells, suggest the following overall hypothesis: pregnancy causes mammary cells that have not mutated to become

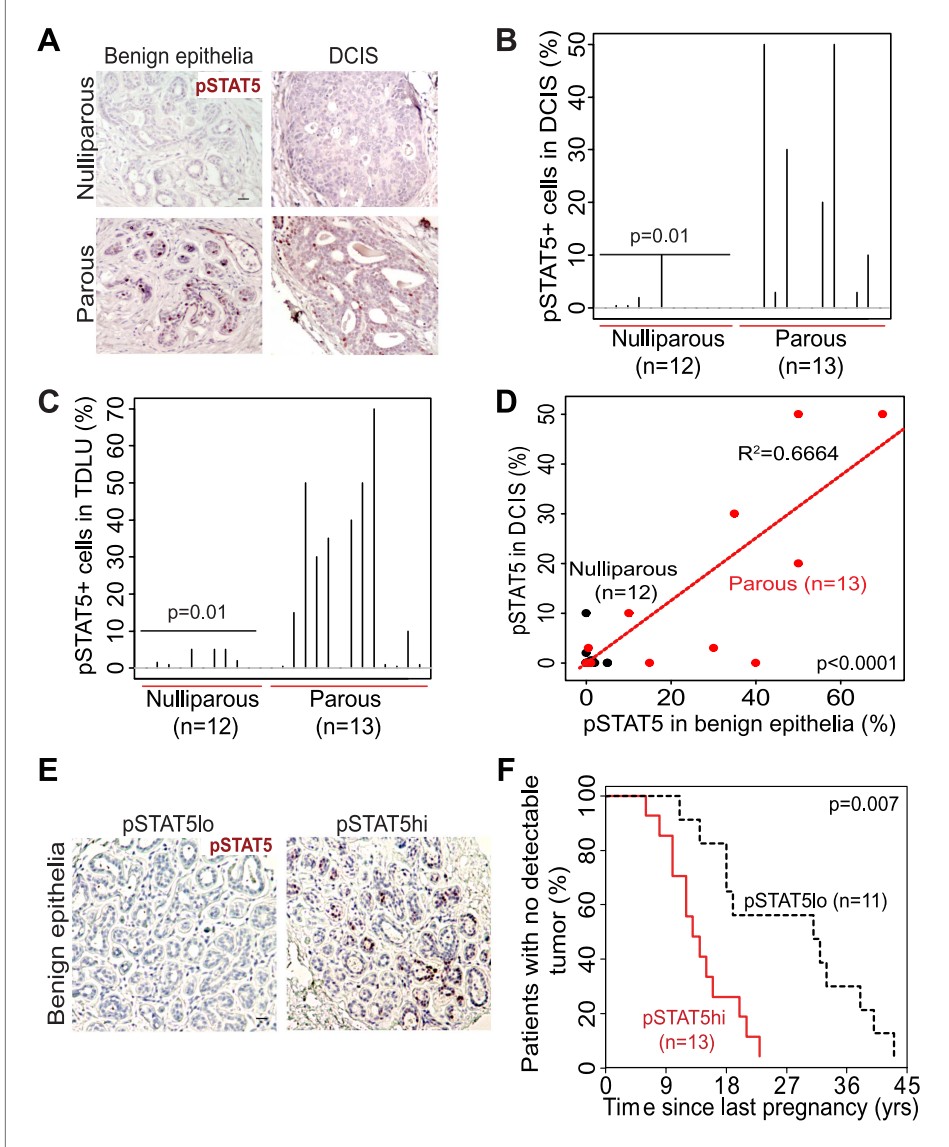

**Figure 7**. STAT5 activation in normal breast and early lesions of women is associated with parity, and pSTAT5 retrospectively predicts decreased intervals between pregnancy and cancer diagnosis. (**A**–**C**) Immunohistochemical staining for pSTAT5 (**A**) and accompanying quantification represented by index plots for DCIS (**B**) and benign breast epithelia (**C**). Scale bar = 20 µm. Pearson's chi-square test determined p values. (**D**) Regression analysis showing a linear correlation between the percentage of pSTAT5 in DCIS and the adjacent benign epithelia in parous (red) and nulliparous (black) women. $R^2$ and p value were derived using a generalized linear regression model. (**E** and **F**) Immunohistochemical staining for pSTAT5 (**E**) in tumor-adjacent breast epithelia stratified patients into pSTAT5low and pSTAT5high. The graph shows the time between the most recent pregnancy and the diagnosis of breast cancer in patients. Age at first pregnancy was included as a confounding factor in analysis of survival curves of women. Generalized Gehan–Wilcoxon test determined p value. Epidemiological characteristics of these patients are presented in *Figure 7—source data 1*.

The following source data are available for figure 7:

**Source data 1**. Descriptive characteristics of patients.

resistant to transformation by a future oncogenic insult, but instigates whichever mammary cells have already mutated to progress to cancer. Therefore, the age-dependent effect of pregnancy on breast cancer risk in women may be due to age-associated gradual accumulation of mutated cells and early lesions which are stimulated by a delayed pregnancy to progress to cancer.

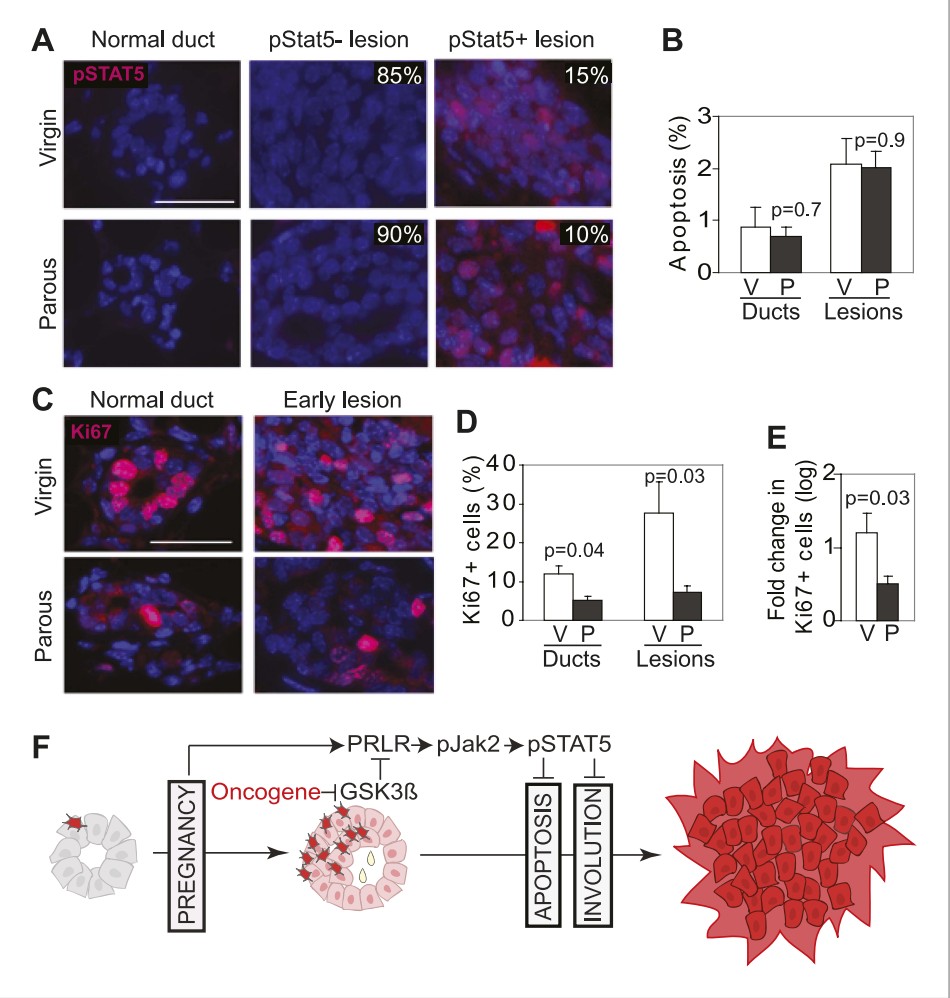

**Figure 8**. Pregnancy reprograms normal mammary cells to resist transformation by a future oncogene. Mice that had completed a pregnancy, 3 weeks of lactation, and 2 months of involution were injected with RCAS-*caErbb2*. 3 weeks later, the resulting lesions were compared with those in age-matched virgin controls for pSTAT5 (**A**), apoptosis (via TUNEL) (**B**), and Ki67 (**C** and **E**). Levels of apoptosis (**B**) and proliferation in normal ducts (**D**) of uninfected mice are shown for comparison. Columns represent the mean, and error bars the SEM. Student's *t* test measured p values. n = 5 mice. (**F**) Schematic Model. Breast cells with oncogenic activation (red) progress to cancer slowly due to the apoptosis anticancer barrier. However, with a pregnancy, these preexisting precancerous cells activate PRLR-Jak2-STAT5 signaling (becoming pink), and maintain the activated state of this pathway even at involution likely through oncoprotein-initiated phosphorylation and inactivation of GSK3β. pSTAT5 overcomes both the apoptosis anticancer barrier and the apoptotic force unleashed by involution, consequently accelerating progression to malignancy.

The following figure supplements are available for figure 8:

**Figure supplement 1**. STAT5 is activated to similar levels by pregnancy and lactation in mammary glands of both young and older mice.

---

Envisioning an application in preventing breast cancer in women having a late age pregnancy who may have already accumulated precancerous cells, we prophylactically treated fully involuted, early lesion-bearing mice with several small molecule inhibitors targeting Jak2 or STAT5 activity. This brief treatment induced apoptosis in early lesions and caused them to regress, and importantly led to significant delay in tumor appearance. Although all mice eventually developed tumors, this significant delay in age of cancer diagnosis is what matters clinically in cancer prevention. Furthermore, our data predict that intermittent treatment with anti-Jak2/STAT5 may inhibit newly formed early lesions and

further delay tumor appearance or may even completely prevent cancer in some of them. pSTAT5 is detected in human breast early lesions, especially in women who have had a pregnancy; therefore, prophylactic therapy targeting Jak2-STAT5 signaling may lower breast cancer risk in high risk women.

This potential clinical implication of our study takes on additional significance because there has been little further success in developing breast cancer chemoprevention since the introduction of inhibitors of estrogen signaling (*Hutchinson, 2011*; *Lyons et al., 2011*). Even for these inhibitors to be effective, women have to take them continuously for several years, sometimes with prolonged side-effects (*Hutchinson, 2011*). The alternative prevention modality indicated by our results may require only a short-term treatment after pregnancy and lactation; thus, it may be more acceptable to women who are as yet breast cancer-free. Additionally, unlike current prevention strategies (*Bode and Dong, 2009*), this alternative regimen may prevent ER-positive cancer as well as other subtypes: ER is negative in 71% of caErbB2-initiated tumors and in 63% of Wnt1-induced tumors in our parous mice although ER is present in the great majority of early lesions induced by either caErbB2 or Wnt1 (data not shown). Therefore, this new chemoprevention strategy may potentially have a significant impact on breast cancer prevention.

pSTAT5 levels are low in early lesions from virgin MMTV-*tva* mice infected by RCAS-*caErbb2*, and not surprisingly, brief treatment with either Jak2 or STAT5 inhibitors had little effect in this very small cohort of mice (*Figure 6*). However, this result should not be interpreted to conclude that STAT5 inhibition has no value in preventing breast cancer in nulliparous mice or humans. Prophylactic treatment with Jak2/STAT5 inhibitors for a longer time, in a large cohort, or in a subgroup selected for high baseline pSTAT5 may very well yield significant preventive benefit. As shown in *Figure 4A*, forced STAT5 activation indeed accelerated tumor formation in virgin mice that were infected by RCAS-*caErbb2*. Some virgin mice had higher baseline levels of pSTAT5 in the normal glands, and their early lesions harbored more pSTAT5$^+$ cells and fewer apoptotic cells and consequently evolved into cancer faster than the early lesions in the virgin mice with lower baseline levels of pSTAT5 (*Figure 4B*). In addition, in any virgin mouse, some of the early lesions had relatively high percentage of pSTAT5 cells, and they correspondingly showed reduced levels of apoptosis (*Figure 4—figure supplement 2*). These observations suggest that even in nulliparous mice or women, increased pSTAT5 in precancerous cells promotes tumorigenesis, and blocking STAT5 activity may also reduce their risk of breast cancer. In support of this notion, we have found that a 2-week-treatment with AG490 caused RCAS-*caErbb2* early lesions in virgin WAP-*tva* mice to deactivate STAT5 (*Figure 6—figure supplement 1C*) and to induce apoptosis (*Figure 6—figure supplement 1D*). Therefore, pSTAT5 promotes early lesion progression to cancer in both nulliparous and parous mice, and chemoprevention targeting Jak2/pSTAT5 may lower the risk of breast cancer in both nulliparous and parous women.

The majority of human premalignant lesions (such as ADH) probably do not progress and do not pose a significant risk, and only 20% of patients with these premalignant lesions in their breasts are eventually diagnosed with breast cancer (*Degnim et al., 2007*). Our data suggest that pSTAT5 in premalignant breast lesions in women may predict a higher chance of progression to clinical cancer, and therefore may help differentiate high risk vs low risk patients and may affect treatment and prevention options. Moreover, increased pSTAT5 even in histopathologically normal breast epithelia may predict a high risk of breast cancer: as discussed in the preceding paragraph, increased baseline levels of pSTAT5 in normal mammary epithelia in nulliparous mice were correlated with higher pSTAT5 levels and lower apoptosis in early lesions of nulliparous mice and predicted an accelerated evolution to overt cancer; pSTAT5 levels were also higher in histologically normal mammary epithelium in MMTV-*Erbb2* mice; and importantly, pSTAT5 was elevated in histologically normal breast epithelia adjacent to DCIS with increased pSTAT5, and high pSTAT5 in tumor-adjacent normal appearing breast epithelia predicted shorter time from the last pregnancy to cancer diagnosis. It is also possible that pSTAT5 levels in normal breast epithelia may serve as a biomarker for estimating the efficacy of anti-Jak2/STAT5 and other chemopreventive strategies. Observation of a larger drop in baseline pSTAT5 levels in breast biopsies following prophylactic chemotherapeutic intervention might indicate a more effective elimination of the threat of potential malignancy.

In conclusion, the results presented in this study identify the stimulatory effect of pregnancy-associated STAT5 activation on cancer initiation in the parous breast, delineating novel potential preventative strategies to combat the increased risk of breast cancer faced by women who have a late-age pregnancy. Further, these data suggest a potential reason for pregnancy's age-dependent effect on breast cancer risk, based on the time at which an oncogenic insult occurs relative to the time at which pregnancy

commences, thereby shedding some light on the dual-role played by pregnancy in human breast cancer.

## Materials and methods

### Experimental animals

MMTV-*tva* (MA) and WAP-*tva* (WA) mice in a FVB genetic background used in these studies have been previously reported (*Du et al., 2006*; *Haricharan et al., 2013*). All animals were euthanized according to the NIH guidelines. The animal protocol was approved by the IACUC of Baylor College of Medicine, Houston, TX.

### Virus preparation and delivery

RCAS virus was prepared as described earlier (*Du et al., 2006*). Briefly, the retrovirus was transfected into DF1 chicken fibroblast cells using Superfect (Qiagen, Gaithersburg, MD) and viral particles were harvested over 1 week. The viral particles were concentrated by ultracentrifugation and frozen for titration and intra-ductal injection. Titration was carried out through limited dilution transduction of DF1 cells. The lentiviral vector (FUCGW) carrying either *GFP* alone or both *GFP* and *caErbb2* was prepared as described earlier (*Bu et al., 2009*).

### Early lesion and tumor studies

4 to 7 days after intraductal injection of RCAS, the experimental group consisting of approximately half the mice was mated. Pups were weaned at lactation day 21. All mice were palpated thrice weekly for tumor incidence. Tumor-free mice were euthanized 7 months (caErbB2/caSTAT5a), 12 months (caErbB2), or 24 months (Wnt-1) post injection. Mammary glands analyzed for the incidence of early lesions were removed from the experimental and control animals either 1.5 weeks (P7.5), 2 weeks (caSTAT5a/caErbB2), 5.5 weeks (L12), 8.5 weeks (I10), or 6 months (Wnt-1) post injection. A small subset of parous animals (<20%) had palpable tumors at the time of weaning. To ensure that the presence of a tumor did not affect pregnancy, lactation, or involution in these mice, we monitored average litter size at the time of weaning and ensured they were comparable between mice that bore tumors at involution day 2 (n = 9) and mice that did not (n = 49).

### Drug treatments

AG490 was administered intraperitoneally based on previous in vivo studies (*Burdelya et al., 2002*). C188-9 was administered at the maximum tolerated daily dose (100 mg/kg BW) based on drug toxicity studies carried out in mice by Dr David J Tweardy. C188-9 was delivered intraperitoneally using Becton–Dickinson LoDose ½ cc U100 insulin syringes. Both drugs were freshly diluted in DMSO before each administration. Ruxolitinib was administered intraperitoneally once a day for 10 days at 100 mg/kg based on a modified protocol from previous in vivo studies (*Quintas-Cardama et al., 2010*).

### Tissue harvest

Tumors and mammary tissue were fixed in 4% paraformaldehyde overnight at 4°C, paraffin-embedded, and sliced into 3-μm sections. The sections were deparaffinized in xylene, rehydrated in graded alcohol, and used for histology and immunostaining. The tumor tissue for Western blotting analysis was snap-frozen in liquid nitrogen and stored at −80°C.

### Whole-mounting of mammary glands

#4 mammary glands were whole-mounted and stained with neutral red as previously described (*Moraes et al., 2007*). Whole-mounted tissue was analyzed for lesions and gland morphology under xylene and then paraffin-embedded and sectioned for immunostaining.

### Immunostaining and microscopy

Immunohistochemistry (IHC) and immunofluorescence (IF) were performed as described earlier (*Du et al., 2006*). Antigen retrieval was carried out by heating sections in 10 mM sodium citrate, pH6.0. MOM and VectaStain Elite ABC Rabbit kits (cat.no. PK-2200 & PK-6101; Vector Labs) were used according to manufacturer's protocols. Primary antibodies used included mouse monoclonal antibody against HA (1:500; Covance), Bcl$_{XL}$ (1:40; Santa Cruz), FLAG (1:500; Sigma), and vWF (1:200; DAKO); rabbit IgG specific for pSTAT5 (which recognizes both pSTAT5a and pSTAT5b) (1:200; Cell Signaling),

cleaved caspase 3 (1:200; Cell Signaling), PRLR (1:500; Santa Cruz), pSTAT3 (1:200; Cell Signaling), STAT5a (1:200; Santa Cruz), pSTAT1 (1:200; Cell Signaling), pHistone3 (1:200; Millipore), and Ki67 (1:200; Novocastra); and goat IgG specific for pSTAT6 (1:50; Santa Cruz). Incubation with the primary antibody for IF staining was overnight at 4°C, while incubation with primary antibody for IHC was 1 hr at room temperature. Nuclei were counterstained with 4'-6-diamidino-2-phenylindole (DAPI)-containing mounting medium and hematoxylin, respectively, for IF and IHC. IHC for pSTAT5 in all human samples was controlled for quality of fixation and paraffin embedding of tissues by scoring the quality of pHistone3 staining of all tissues that were negative for pSTAT5. Only those samples that passed this quality control test were included in the final analysis. Bright field images were captured using a Leica DMLB microscope, and images were processed with Magnavision and Adobe Photoshop software. Fluorescent images were captured with a Zeiss Axiskop2 plus microscope. Images were processed with Axiovision and Adobe Photoshop software.

## Terminal deoxynucleotidyl transferase dUTP nick-end labeling assay

Paraffin-embedded gland and tumor sections were treated in proteinase K and subjected to the terminal deoxynucleotidyl transferase dUTP nick-end labeling (TUNEL) assay using the ApopTag Red in situ TUNEL detection kit (Chemicon, S7165). Nuclei were counterstained with DAPI-containing mounting medium.

## Quantification of stained sections

For tumor samples, five random fields were viewed and a total of at least 5000 cells were counted per sample. For quantification of apoptotic, proliferative, pSTAT5+, PRLR+, and p53+ cells in early lesions, at least 2000 cells were counted per sample. For quantification of apoptotic and proliferative cells in normal ducts, at least 10 ducts were counted per section, and a total of at least 1000 cells were counted per sample. When co-staining was impractical, consecutive sections were cut from formalin-fix and paraffin-embedded (FFPE) tissue and used for staining. ImageJ software was used for counting, and either DAPI or hematoxylin nuclear staining was used to identify the total number of cells. ImageJ software was also used to quantify the total percentage of HA+ area in the entire mammary gland from sequential bright-field pictures taken at 2.5 × resolution. Fixed thresholds were set to analyze both experimental and control mammary glands.

## Western blotting analysis of mammary tumors

Protein was extracted from frozen tissue, and lysates analyzed as previously described (*Li et al., 2001*). Primary antibodies used were mouse monoclonal HA (1:1000; Covance), BclXL (1:500; Santa Cruz), cyclinD1 (Cell Signaling, 1:500), and STAT5b (Santa Cruz; 1:500); and rabbit polyclonal Bcl2 (1:500; Santa Cruz), PRLR (1:500; Santa Cruz), pSTAT5 (1:500; Cell Signaling), STAT5a (1:500; Santa Cruz), and GAPDH (1:2000; Santa Cruz). Secondary antibodies used were HRP-conjugated goat anti-mouse and anti-rabbit (1:5000; Pierce). Super-signal Femto-Chemiluminescence substrate (Thermo Scientific) was used to visualize bands on Western blots.

## Human cohort analysis

Only 10/27 women in cohort 1 opted for BRCA testing and all 10 were BRCA negative. The remaining 17 have undetermined BRCA status. For cohort 2, however, all women were tested for BRCA status, and only 2/24 were found positive.

## Statistical analysis

All numbers in the text were represented as mean ± standard error of the mean. Statistical analysis of quantification of stained sections was done using ANOVA or Student's *t* test for independent samples with Holm's correction for multiple comparisons when distribution of data was judged to be normal. For hypothesis confirmation in *Figure 6—figure supplement 1B* Student's one-sided *t* test was used. Where distribution was not normal (assessed using Q–Q plots with the Wilk-Shapiro test of normality), either Kruskal-Wallis or Wilcoxon's Rank Sum test was used. In all cases, at least 10 lesions were quantified per mouse. Holm's correction was also used where required when using non-parametric tests. For categorical data with <15 data points in each group, the Fisher's Exact test was used. For categorical data with ≥15 data points in each group, the Pearson's chi-square test was used. Tumor-free survival analysis was done using the Generalized Gehan-Wilcoxon test with rho = 1, and Kaplan-Meier survival curves were generated in R. The regression equation for correlation between pSTAT5 and apoptosis

was configured to include lesion size (total number of cells in the lesion) along with pSTAT5% as potential predictive factors of apoptosis, and only pertains to lesions from a single mouse (red). All graphs and regression analyses were generated either in MS Excel or R.

## Acknowledgements

We thank Tammy Tong for technical assistance and Daniel Medina, Jeffrey Rosen, Michael Lewis, Adrian Lee, Hoang Nguyen, Sendurai Mani, Xiang Zhang, Gary Chamness, and Matthew Bainbridge for stimulating discussions and/or critical review of this manuscript.

## Additional information

### Funding

| Funder | Grant reference number | Author |
|---|---|---|
| Congressionally Directed Medical Research Programs | BC073703, BC085050, BC112704 | Yi Li |
| National Institutes of Health | CA113869, CA124820, U54CA149196 | Yi Li |
| Nancy Owens Memorial Foundation | | Yi Li |
| Dan L Duncan Cancer Center | P30CA125123 | Yi Li |
| Lester and Sue Smith Breast Center | P50CA058183 | Yi Li |
| Specialized Programs of Research Excellence | P50CA058183 | Jie Dong |
| Huffington Center | T32AG000183 | Kimberly Holloway |
| Congressionally Directed Medical Research Programs | BC083190 | Jay P Reddy |
| National Institutes of Health | CA149783, CA153659 | David J Tweardy |
| Cancer Prevention Research Institute of Texas | RP100421 | David J Tweardy |
| Robert and Janice McNair Foundation | | Jay P Reddy |
| Cancer Prevention Research Institute of Texas | RP101499 | Sarah Hein |

The funders had no role in study design, data collection and interpretation, or the decision to submit the work for publication.

### Author contributions

SH, JD, Conception and design, Acquisition of data, Analysis and interpretation of data, Drafting or revising the article; SH, KH, Acquisition of data, Drafting or revising the article; JPR, ZD, MT, RA, WW, Acquisition of data, Contributed unpublished essential data or reagents; SGH, Analysis and interpretation of data, Drafting or revising the article; SH, YL, Conception and design, Analysis and interpretation of data, Drafting or revising the article; SJ, VFB, CG, HZ, Acquisition of data, Analysis and interpretation of data; PJS, Acquisition of data, Analysis and interpretation of data, Drafting or revising the article; CKO, Conception and design, Analysis and interpretation of data; DJT, Conception and design, Acquisition of data, Drafting or revising the article, Contributed unpublished essential data or reagents

### Ethics

Animal experimentation: All animals were euthanized strictly according to the Guide for the Care and Use of Laboratory Animals of the NIH and every effort was made to only include the number of mice required for accurate statistical analysis and to minimize suffering. All surgery was performed under either isoflurane or ketamine. The animal protocol (#AN2834) was approved by the IACUC of Baylor College of Medicine, Houston, TX.

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
