## [Decision Letter]

Thank you for sending your work entitled “Mechanism and Preclinical Prevention of Increased Breast Cancer Risk Caused by Pregnancy” for consideration at *eLife*. Your article has been favorably evaluated by a Senior editor and 3 reviewers, one of whom is a member of our Board of Reviewing Editors.

The Reviewing editor and the other reviewers discussed their comments before we reached this decision, and the Reviewing editor has assembled the following comments to help you prepare a revised submission.

The manuscript “Mechanism and Preclinical Prevention of Increased Breast Cancer Risk Caused by Pregnancy” provides an exhaustive study potentially linking breast-cancer risk associated with “late”, pregnancy with STAT5 activation in breast tissue following pregnancy. The murine experimental model used for the study involves introduction of oncogenes like ERBB2 and WNT1 through RCAS viral infection in 5-7 week old mice, impregnated <=1 week later, followed by assessment of the effect of pregnancy on premalignant lesions at weeks 1.5, 5.5 and 8.5 post infection (pregnancy day 7.5, lactation day 12, and involution day 10). Through a series of cleverly crafted experiments, the authors propose that STAT5 activation during pregnancy causes apoptosis evasion in premalignant lesions that accumulate in the breast with age. The study concludes with experiments to show that anti-STAT5 treatment post pregnancy could lower breast cancer risk.

The layout of experiments is logical and appropriate controls are used. The manuscript is very well written as well. However, there are several major concerns that must be addressed before we would consider accepting the manuscript.

1) The opening premise of the study is that “While a first pregnancy before age 22 lowers breast cancer risk, a pregnancy after age significantly increases life-long breast cancer risk”, but the entire experimental model system used here compares the progression of oncogenic lesions in virgin versus 5-7 week young mice, presumably to “keep ageing-associated variables constant”. To address the concept of “long term”/ “life-long” breast cancer risk following late age pregnancy, the authors should see if the course of STAT5 activation following “early” versus “late” pregnancy is different, qualitatively or quantitatively.

2) The authors state “among the STATs known to be activated in the mammary gland, STAT5 is the predominant family member activated in parous early lesions (data not shown)”. This is an important claim and the data should be shown. This statement is not necessarily obvious in the literature, as STAT3 has been shown to be a potent tumor protein in breast cancer (Garcia et al., Constitutive activation of Stat3 by the Src and JAK tyrosine kinases participates in growth regulation of human breast carcinoma cells, Oncogene (2001) 20: 2499; Gritsko et al., Persistent Activation of Stat3 Signaling Induces Survivin Gene Expression and Confers Resistance to Apoptosis in Human Breast Cancer Cells, Clin Cancer Res 2006 12: 11). Based on the observations here, do the authors think pSTAT5 status could serve as a biomarker for efficacy of chemopreventive agents?

3) Would Jak2 inhibitor decrease the tumor onset of virgin WAP-*tva* mice infected with caErbB2?

4) Figure 7 finds high phospho-stat5 in adjacent, histologically normal area, but this is not the case in the mouse model data shown. This is a concern because it raises the question if mouse recapitulates the p-stat5 biology seen in human breast. One possible experiment is to use GEM models of ErbB2-induced breast cancers. These models develop tumors in a stochastic manner after long latency. One may ask if early pregnancy increases the percentage of STAT5+ve cells in epithelial cells, irrespective of their hyperplastic status.

5) It is my understanding that PRLR is usually localized to the cell membrane; the nuclear stain showed in Figure 3 is confusing. How did the authors rule out a possible non-specific staining?

6) It is not clear the use of C188-9, a *Stat3* inhibitor, provides support for the authors' findings on the role played by *Stat5* activation.

7) The authors overstate the clinical impact of their work in resolving the paradox of pregnancy-induced breast cancer protection versus progression depending on age. The authors should tone down their interpretation and focus only on the direct impact of their findings.

---

## [Author Response]

*1) The opening premise of the study is that “While a first pregnancy before age 22 lowers breast cancer risk, a pregnancy after age significantly increases life-long breast cancer risk”, but the entire experimental model system used here compares the progression of oncogenic lesions in virgin versus 5-7 week young mice, presumably to “keep ageing-associated variables constant”. To address the concept of “long term”/ “life-long” breast cancer risk following late age pregnancy, the authors should see if the course of STAT5 activation following “early” versus “late” pregnancy is different, qualitatively or quantitatively*.

We appreciate the reviewers’ understanding of our experimental approach. Per the reviewers’ suggestion, we have quantitatively compared STAT5 activation between an older age pregnancy (16 weeks) and a young age pregnancy (6 weeks) using Western blotting. We find comparable levels of STAT5 phosphorylation (p=0.5) and total STAT5a (p=0.2) in the mammary gland at lactation day 12 regardless of whether the pregnancy is “early” or “late” (Figure 8—figure supplement 1 of the revised submission). These data indicate that STAT5 activation following “early” versus “late” pregnancy is quantitatively comparable, indicating that the results described in the manuscript will likely hold true irrespective of age.

*2) The authors state “among the STATs known to be activated in the mammary gland, STAT5 is the predominant family member activated in parous early lesions (data not shown)”. This is an important claim and the data should be shown. This statement is not necessarily obvious in the literature, as STAT3 has been shown to be a potent tumor protein in breast cancer (Garcia et al., Constitutive activation of Stat3 by the Src and JAK tyrosine kinases participates in growth regulation of human breast carcinoma cells, Oncogene (2001) 20: 2499; Gritsko et al., Persistent Activation of Stat3 Signaling Induces Survivin Gene Expression and Confers Resistance to Apoptosis in Human Breast Cancer Cells, Clin Cancer Res 2006 12: 11). Based on the observations here, do the authors think pSTAT5 status could serve as a biomarker for efficacy of chemopreventive agents*?

We have now included the representative figures and quantification as Figure 3—figure supplement 3, Figure 5—figure supplement 2, Figure 6—figure supplement 1, and Figure 6—figure supplement 3 in the revised submission. Based on our data, we suggest that persistent pSTAT5 can serve as a biomarker for unsuccessful chemoprevention. If STAT5 activation in premalignant cells increases or remains the same after exposure to chemopreventive agents, this might well indicate that the strategy was not effective in the patient. But the most important implication of our findings would be to use STAT5 as the target for chemoprevention, in which case it would also serve as the optimal biomarker. We have included a short discussion on this implication of our results in the Discussion section of the revised manuscript.

*3) Would Jak2 inhibitor decrease the tumor onset of virgin WAP-tva mice infected with caErbB2*?

We have treated RCAS-*caErbb2*-infected virgin WAP-*tva* mice with AG490. We found that AG490 does inhibit STAT5 phosphorylation in *caErbb2*-induced early lesions and also significantly induces apoptosis. These results are now included in Figure 6—figure supplement 1 and are referred to in the text. These data indicate that a Jak signaling inhibitor can prevent early stages of tumor formation in virgin WAP-*tva* mice and therefore, would decrease tumor onset. We will in the future treat a larger cohort of virgin WAP-*tva* mice with a Jak2 inhibitor and confirm that tumor appearance is also delayed. But that will take a much longer time than we can achieve for this revision.

*4)*
Figure 7
*finds high phospho-stat5 in adjacent, histologically normal area, but this is not the case in the mouse model data shown. This is a concern because it raises the question if mouse recapitulates the p-stat5 biology seen in human breast. One possible experiment is to use GEM models of ErbB2-induced breast cancers. These models develop tumors in a stochastic manner after long latency. One may ask if early pregnancy increases the percentage of STAT5+ve cells in epithelial cells, irrespective of their hyperplastic status*.

In the mouse data shown, pSTAT5 levels in normal cells were quantified from the completely uninjected and therefore, completely normal contralateral breast. In our human tissues, on the other hand, the “normal” tissue was tumor-adjacent and several studies have reported that tumor-adjacent histopathologically “normal” tissue is far from normal based on genetic and epigenetic changes (Cho et al., 2010, Anticanc Res; Yan et al, 2006, Clin Canc Res). Moreover, as the oncogene in our mouse model system was tagged and therefore visible through immunofluorescence, it was easy to distinguish premalignant cells from completely normal cells independent of histopathology in our mice. In contrast, in our human tissue, the “normal” cells were determined pathologically in the absence of any molecular indications of normality. These discrepancies between the mouse experiments and human data are likely responsible for the difference in pSTAT5 levels of normal breast cells in the two sets of experiments.

To further address the issue of comparability between the human tissue and mouse experimental data, we have followed the reviewers’ suggestion and used MMTV-*Neu* mice to test whether pregnancy increases the percentage of pSTAT5-positive epithelial cells irrespective of histopathological attributes (see Figure 2—figure supplement 2 of the revised submission). We find that there is no detectable pSTAT5 positivity in histopathologically “normal” mammary epithelial cells from 5 virgin MMTV-*Neu* mice whereas 4/5 MMTV-*Neu* mice at involution day 10 (age-matched) had detectable pSTAT5 in a significant proportion of “histologically normal” mammary epithelial cells. These data recapitulate our findings in human tissue.

*5) It is my understanding that PRLR is usually localized to the cell membrane; the nuclear stain showed in*
Figure 3
*is confusing. How did the authors rule out a possible non-specific staining*?

This is a valid concern. In all our immunohistochemical analyses of PRLR in mouse mammary tissue, we used PRLR-deficient mammary glands as a negative control to test the specificity of the staining. This negative control is from knockout mice which show depleted PRLR mRNA based on qPCR and are not responsive to prolactin stimulation. A detailed characterization of this strain of mouse is being prepared for publication by my colleague Dr. Michael Lewis at Baylor College of Medicine. Although we cannot show this negative control in our manuscript at this time, we are including it here to further confirm the specificity of our immunoassay.Author response image 1.

In Figure 3 of the original manuscript, we showed quantification of total PRLR^+^ cells, but in Figure 3—figure supplement 4 of this revised submission, we also quantified membrane PRLR^+^ cells in the lesions of parous vs virgin mice and found that the increased PRLR-positivity in the parous lesions remained significant (p=0.01).

*6) It is not clear the use of C188-9, a* Stat3 *inhibitor, provides support for the authors' findings on the role played by* Stat5 *activation*.

While C188 is known to inhibit STAT3 activation as shown by our collaborator, Dr. David Tweardy, C188-9 is a derivative of this drug which has comparably potent inhibitory effects on both STAT5 and STAT3 as shown in Figure 6—figure supplement 2 of the revised submission. Since pSTAT5^+^ cells occur at 10-fold higher frequency than pSTAT3^+^ cells in early lesions from our mouse model (18% vs 1.8%), C188-9 is effectively a STAT5 inhibitor in this setting. In accord, the number of pSTAT5^+^ parous early lesions decreases significantly on C188-9 treatment (Figure 6—figure supplement 3) while the low number of pSTAT3^+^ cells in these early lesions remains constant even on treatment with C188-9 (Figure 6—figure supplement 3). Therefore, the use of C188-9 provides an independent confirmation of the necessity for STAT5 activation to mediate pregnancy’s promotion of tumor formation, alongside the experiments using STAT5a^-/-^ mice (Figure 5) and those using two independent Jak inhibitors (AG490 and ruxolitinib; Figure 6).

*7) The authors overstate the clinical impact of their work in resolving the paradox of pregnancy-induced breast cancer protection versus progression depending on age. The authors should tone down their interpretation and focus only on the direct impact of their findings*.

We have now modified the text of the Discussion to focus on the direct impact of our findings.